# Navigating Uncertainty with Compassion: Healthcare Assistants’ Reflections on Balancing COVID-19 and Routine Care through Adversity

**DOI:** 10.3390/healthcare12151544

**Published:** 2024-08-05

**Authors:** Alice Yip, Jeff Yip, Zoe Tsui, Graeme Drummond Smith

**Affiliations:** 1S.K. Yee School of Health Sciences, Saint Francis University, Hong Kong, China; ztsui@sfu.edu.hk (Z.T.); gsmith@sfu.edu.hk (G.D.S.); 2Hong Kong Institute of Paramedicine, Hong Kong, China; jeffreyycyip@gmail.com

**Keywords:** care, COVID-19, coronavirus disease 2019 pandemic, resilience, self-efficacy, healthcare assistants, workforce issues

## Abstract

The coronavirus disease 2019 pandemic created unprecedented challenges for healthcare systems around the world. Healthcare assistants played a vital role in the provision of frontline patient care during this crisis. Despite their important contribution, there exists limited research that specifically examines the healthcare assistant’s experiences and perspectives of care provision during the COVID-19 pandemic. This study explored healthcare assistants’ caring experiences and perspectives on resilience and self-efficacy during the COVID-19 pandemic in Hong Kong. A qualitative descriptive study with semi-structured interviews was conducted with 25 healthcare assistants from public hospitals. Interview recordings were analyzed using thematic analysis. Five main themes emerged from the data: frontline reinforcement: supporting HCAs through resourcing and education amidst the COVID-19 crisis, confronting uncertainty: building personal fortitude in the face of the COVID-19 pandemic, fostering collective resilience through shared support, self-efficacy as a catalyst for adaptive growth, and paving the way for transformation. These findings advocate for the resilience and self-efficacy of healthcare assistants; this may potentially strengthen healthcare system preparedness for navigating unpredictable challenges in the future.

## 1. Introduction

The coronavirus disease 2019 (COVID-19) pandemic has been responsible for over 664 million confirmed cases and 6.7 million deaths worldwide according to the World Health Organization [1]. In the United States, COVID-19 has resulted in a nationwide public health crisis with over 103 million diagnosed cases and approaching 1 million deaths. In Hong Kong, a densely populated metropolitan area of over 7 million residents, COVID-19 has also imposed severe public health and economic tolls, with over one million documented cases and 9000 deaths as of January 2023 [2,3]. Hong Kong’s healthcare infrastructure has been strained to the limits, reminiscent of the 2003 outbreak of severe acute respiratory syndrome (SARS), which exposed vulnerabilities in emerging infectious disease preparedness [4]. At the beginning of the pandemic, healthcare systems worldwide were inadequately prepared for the inflow of patients with COVID-19 [5,6,7,8]. The pandemic has also brought profound adverse impacts on the lives and well-being of much of Hong Kong’s population [9].

Globally, the emergence of the COVID-19 pandemic precipitated a swift transformation of healthcare practice. Healthcare facilities faced a dramatic increase in the volume of patient numbers whilst possessing limited resources to provide services to an unfamiliar demographic [10,11,12,13]. Within these settings, healthcare assistants (HCAs), who dedicate a substantial amount of time to delivering direct patient care, are important to care quality and delivery [14,15]. However, studies suggest HCAs were frequently deputized, their skills not fully leveraged, and their contributions overlooked during the pandemic [16,17,18].

Healthcare assistants are vital members of the healthcare team, as their work has been shown to improve patient comfort during hospitalization [14,15,16]. They provide hospital-based patients with physical care while giving a certain degree of basic psychological and existential support, acting as observant monitors, and quickly alerting nursing clinicians about health status fluctuations or deterioration [17,18].

In the realm of healthcare, resilience delineates the capacity of healthcare workers to effectively manage and adapt to the demands and pressures of their work environment. Resilient healthcare workers can support their psychological and emotional health while implementing up-to-standard patient care, even when faced with challenging situations [19,20]. Self-efficacy, defined as a HCA’s belief in their own capabilities to perform actions and achieve outcomes, has been thought to have a direct bearing on individual performance in particular contexts [21,22]. Although HCAs take on physically strenuous tasks, they are often seen as low-skilled and mundane, frequently encountering disrespect and mistreatment [17,18,23]. Exclusion from important conversations and limited opportunities for career growth and advancement can lead to HCAs feeling stuck in their occupational role without the prospect of professional progression. This potential lack of mobility and perpetual perception of being ‘bottom of the totem pole’ can cause HCAs to experience a lack of purpose, loss of motivation, and emotional exhaustion—all key factors contributing to burnout [17,23,24,25]. However, over the past two decades, HCA roles have expanded and gained greater value [18,26,27]. Enhanced self-efficacy has eventually brought about an increased level of HCA involvement across all patient care aspects, including planning [17,18,28,29,30]. These developments have had a positive effect on job satisfaction, retention inclination, and a reduction in the probability of resignation [31,32,33,34,35,36].

During the pandemic, which rapidly transformed healthcare operations, having a resilient and skilled HCA workforce appeared critical for healthcare systems to meet escalating care demands. Specifically, HCAs needed to adapt to quickly evolving protocols, take on extended responsibilities, and provide safe and up-to-date patient care during intensely stressful and uncertain conditions [17,18,37]. Retaining empowered and engaged HCAs within the workforce ensured continuity of care and avoided staffing shortfalls that would further burden the limited available resources [38]. Empowered HCAs could adapt to changing protocols for patient care during this stressful time [39]. Alongside professional clinicians, HCAs represent an important, resilient workforce that can adapt to unpredictable situations and provide consistent and safe high-quality patient care.

The International Council of Nurses (ICN) recognizes that HCAs were an indispensable yet disproportionately impacted segment of the healthcare workforce during the COVID-19 pandemic. However, data on infections and deaths among HCAs are severely limited across most nations, highlighting a significant gap in understanding the full extent of their experiences and challenges. The ICN advocated standardizing data collection on cases and fatalities among HCAs, which is crucial to understanding and mitigating the occupational risks they face [40]. While some studies have investigated the impact of the pandemic on healthcare workers in general, limited qualitative research has specifically explored the unique perspectives and experiences of HCAs themselves, particularly during times of crisis, such as those seen during the COVID-19 pandemic [41]. This study aimed to address this gap by focusing exclusively on the voices of HCAs, providing valuable insights into their resilience and self-efficacy during this unprecedented time. The findings of this study will provide a greater understanding of HCA perspectives, addressing gaps in knowledge about the development of resilience and self-efficacy of HCAs both within the localized context of Hong Kong and the broader international landscape. By delving into the specific challenges, coping mechanisms, and support systems of HCAs in Hong Kong, this study offers a unique contribution to the existing literature, which has largely overlooked this critical workforce.

The study aimed to explore HCAs’ experiences and perspectives on resilience and self-efficacy during the COVID-19 pandemic in Hong Kong. Based on the findings of this study, management of the healthcare system can identify ways to better support and empower this critical workforce during future public health crises. By understanding the specific needs and challenges faced by HCAs, healthcare organizations can develop targeted interventions and policies to enhance their resilience and self-efficacy, ultimately strengthening the healthcare system’s response to future pandemics or other crises.

## 2. Methods

### 2.1. Study Design

This descriptive qualitative study adopted an eclectic, flexible approach grounded in constructivist inquiry [42,43]. The aim was to generate rich, realistic findings that reflect the experiences of HCAs caring for COVID-19 patients during the pandemic. The researchers opted for an adaptive methodology without stringent rules to optimize the data collection and analysis based on the specific research goals. This enabled tailored methods to prioritize eliciting detailed accounts from participants. This study involved semi-structured interviews with HCA participants and a thematic analysis of the interview transcripts (HCAs in their own words) to systematically identify major themes related to their perspectives and experiences during the COVID-19 pandemic [44]. The Consolidated Criteria for Reporting Qualitative Research (COREQ) checklist was applied in this study as a framework to ensure rigor in reporting the qualitative methods and findings [45].

A local academic institution’s research and ethical committee granted ethical approval (HRE210101). Participation was entirely voluntary, participants retained the right to withdraw without penalty, and informed consent was obtained after a comprehensive explanation of the study’s purpose, procedure, and potential risks and benefits. They were informed that their experiences and the content of their interviews would be shared in research dissemination and their anonymity was guaranteed to protect their personal identity.

### 2.2. Participants

Employing a purposive sampling approach, 25 HCAs from public hospitals in Hong Kong who actively attended to COVID-19 patients were selected for study participation. The HCAs were primarily identified through the nursing personnel’s social networks, including colleagues and hospital managers. Participants were recruited from three hospitals with existing COVID-19 isolation wards. The research team informed the managers of these hospitals about the study and expressed an interest in interviewing their HCAs. HCAs who showed an interest in participating were then contacted individually by the research team for follow-up. Their eligibility was assessed based on the following criteria: (i) ≥18 years of age and (ii) at least three months of clinical exposure to individuals diagnosed with COVID-19, including both continuous and intermittent periods of active care.

### 2.3. Data

All interviews were conducted in Chinese by the second author over Zoom video-conferencing within a private, password-protected virtual space from March 2021 to December 2021. Interview appointments were scheduled at the convenience of each participant. An interview guide was utilized across all sessions, with durations ranging from 30 to 60 min to ensure consistency in the data collection process. Study data were only accessible to the research team. The interview questions were validated by an expert panel comprising a qualitative research scholar and a psychological consulting specialist to confirm alignment with the aim and focus of this research study, as shown in Table 1. The questions were centered on the participants’ psychological experiences and emotional responses when undertaking the challenging responsibility of providing clinical care to patients during the COVID-19 pandemic. The inquiries probed changes to work duties, personal lives, coping strategies employed, insights gained, and perspectives on their critical role in preventing the spread of COVID-19. Data saturation was reached when the gathering of more data did not yield any new insights, themes, or concepts related to this study. Therefore, data saturation was attained during the 23rd interview, after which no new insights emerged within the final two interviews, and no participant withdrew during data collection, indicating cessation of the data collection.

All interviews were audio-recorded and supplemented by field notes taken concurrently to contextualize the sessions and assist subsequent analysis. Psychological support was made available by the research team if any participant exhibited emotional distress, so as to mitigate further harm. The participants were also notified of their right to voluntarily withdraw if desired.

### 2.4. Analysis Strategy

Following the data collection process, thematic analysis was performed using NVivo software (NVivo Version 12, QSR International, Burlington, MA, USA) without delay in analyzing the interview transcripts [46]. Two researchers (A.Y. and Z.T.) created verbatim transcripts of the audio recordings and then translated the transcripts into English. Back-translation was conducted by another researcher (J.Y.) and a professional translator was consulted to further ensure semantic equivalence. The first author (A.Y.), as the primary coder, initiated the coding and evaluated preliminary findings at weekly team meetings, where feedback was provided on coding and categorization until a consensus was reached and the codebook finalized.

### 2.5. Rigor and Trustworthiness

To ensure accuracy, the researcher (J.Y.) independently double-coded the transcripts. Several strategies were established in this study to ensure the trustworthiness of the findings, including member checking after the interviews, auditing the data analysis process, engaging in reflexive discussion among the researchers to assess participants’ psychological feelings, and purposeful sampling of HCAs from multiple wards to capture extensive perspectives and experiences [47,48,49,50,51]. The final themes were derived through consensus and are detailed in the results.

## 3. Results

Twenty-five HCAs were invited to undertake qualitative interviews about their involvement in delivering care throughout the COVID-19 pandemic. The demographic and participants’ characteristics are delineated in Table 2. Regardless of the wards assigned by their working organizations, they all provided healthcare services to patients diagnosed with COVID-19. The analysis resulted in the conceptualization of resilience and self-efficacy and the emergence of five themes, as shown in Table 3.

### 3.1. Theme 1: Frontline Reinforcement: Supporting HCAs through Resourcing and Education Amidst the COVID-19 Crisis

The provision of adequate staffing, supplies, training, and transparent communication was critical to help mitigate apprehension and bolster preparedness amongst HCAs during the COVID-19 pandemic. However, substantial gaps were exposed in organizational support for this frontline workforce at the onset of the public health crisis. Persistent shortfalls in personal protective equipment (PPE) heightened the perceived occupational transmission risks for HCAs, though proactive self-education helped compensate for insufficient formal training. Frequent policy changes also posed adaptation challenges for HCAs despite having a strong motivation to remain up-to-date on evolving best practices. Targeted investments to reinforce human, physical, and educational resources for HCAs may foster greater resilience and safety as healthcare systems continue navigating COVID-19 challenges.

#### 3.1.1. Heightened Apprehension Surrounding Perceived Infection Risks

The abrupt arrival of the highly contagious SARS-CoV-2 virus precipitated intense anxiety and distress among HCAs regarding the perceived transmission risks, compounded by PPE scarcity. One HCA described the following:
*“Working on the frontlines without enough masks, gloves, and gowns puts us all in danger. It feels like we’re being treated as expendable.”*(P04, female)

Despite assurances from hospital leadership of expected PPE resupply, acute shortages of N95 respirators, surgical masks, gloves, and isolation gowns persisted for several months. This unrelenting lack of adequate protective supplies prompted consideration of occupational resignation or refusal to care for COVID-19 patients by some HCAs upon the pandemic onset. Those stationed in high-risk wards, including emergency departments and intensive care units, voiced the greatest trepidation given their elevated viral exposure. However, the timely provision of sufficient PPE eventually alleviated most apprehensions. Another HCA described the following:
*“When news broke of COVID-19 reaching our city, I was incredibly anxious about having adequate PPE to protect myself. Our supervisors kept reassuring us supplies were on the way, but the delays in receiving them made me seriously consider resigning.”*(P10, female)

#### 3.1.2. Proactive Self-Education on Infection Prevention Protocols

Given the limited time available for formal training on the rapidly evolving COVID-19 protocols, HCAs displayed commendable initiative in self-educating on transmission prevention outside of work hours. Through independent online research and knowledge sharing amongst colleagues, they became well-versed in the latest institutional policies and best practices endorsed by public health authorities. Reinforcing infection control knowledge through informal networks helped compensate for perceived training deficiencies and boosted HCAs’ confidence in adhering to complex, frequently changing guidelines. However, concerns were raised regarding differences in access to technology and reliable information amongst HCAs. Targeted efforts to provide both digital and print educational resources could further support ongoing workforce development. Two participants explained this:
*“Our training was limited, so I took it upon myself to thoroughly read the updated guidelines that were emailed by our infection control team. I asked the nurses to help me print copies to share with co-workers who have been off for several days, so they can review the key points posted on the wall in the pantry when they return.”*(P15, female)
*“I searched through online video recording from ‘iLearn’ nightly for the latest details on proper hand hygiene and updates to isolation precaution policies. Staying updated gave me more confidence in safely performing my duties. This showed how I was able to adapt and learn.”*(P23, male)

#### 3.1.3. Challenges Adjusting to Evolving Practices

The continual modifications of COVID-19 policies posed numerous adaptation challenges for HCAs striving to deliver safe, high-quality care. Frustrations were commonplace regarding insufficient communication and guidance around new procedures, from visiting restrictions to isolation protocols. One participant reported the following:
*“Just as I became comfortable with one set of COVID-19 protocols, they would get updated again. It was incredibly frustrating to re-learn processes with minimal guidance.”*(P09, female)

Despite a strong motivation to remain updated, the cognitive load from perpetual protocol changes was mentally taxing for many. One participant described the following:
*“Trying to keep up with the constantly changing rules for isolation gown use, cleaning routines, visitor policies—it made my job exponentially more difficult.”*(P12, female)

### 3.2. Theme 2: Confronting Uncertainty: Building Personal Fortitude in the Face of the COVID-19 Pandemic

The participants expressed that gaining knowledge through preparation, training, and practice was fundamental in reducing the uncertainty they felt about caring for COVID-19 patients. This process of acquiring expertise influenced how the HCAs perceived the repetition of daily tasks, bridging skill gaps, and building self-confidence. The healthcare workers together had to quickly adapt to new infection control procedures and treat an unfamiliar virus. Through repetitively performing safety rituals, soliciting on-the-job guidance, and gaining hands-on experience, they gradually became more assured in their capabilities. The repetitive nature of donning PPE, monitoring patients, and adhering to guidelines slowly transformed their initial hesitations into confidence. Watching experienced colleagues model techniques helped them to learn and improve their competencies. Each shift enabled them to strengthen their skills and overcome gaps through practice.

#### 3.2.1. Fostering Routines through Repetition

The training and practical knowledge provided via leadership directly impacted how the HCAs established daily rituals in their work. HCAs had to absorb varied information and hands-on training techniques and apply them when caring for COVID-19 patients. They conveyed frustration, anxiety, fear, and a desire for guidance when discussing how the knowledge they had acquired influenced their care for these unique patients. One HCA described the following:
*“Performing the same tasks shift after shift has allowed me to develop set procedures that foster a sense of stability despite the uncertainty.”*(P23, male)

The repetitive nature of donning protective gear, conducting vital sign measurements, and adhering to safety protocols gradually transformed their initial hesitations into ingrained routines. One HCA reported the following:
*“The constant repetition of the same safety steps has slowly boosted my confidence. What once seemed daunting is now familiar.”*(P06, male)

Through the persistent repetition of key tasks, the HCAs were able to establish a reliable set of protocols to follow. This transformation of doubt into routine care was critical for being able to focus fully on providing attentive care to patients. One HCA described the following:
*“I follow the same routine day after day, anxious about potentially contracting the virus myself. However, as I care for each patient, my fears fade, and I am filled with empathy.”*(P01, female)

#### 3.2.2. Building Self-Confidence through Mastering Skills

The HCAs had to rely on their own initiative when carrying out infection control procedures during daily work. Some would strictly adhere to the posted guidelines for donning PPE. Through repetitive and successful demonstrations of these safety protocols, the HCAs developed an increased sense of self-confidence in their skills. One HCA described the following:
*“At first, I felt unsure caring for COVID-19 patients. But with each shift, I gained more confidence in my skills and ability to protect myself and them.”*(P08, female)

The healthcare workers’ repetition of safety protocols, until they became automatic routines, allowed them to gain self-assurance and bridge any skill gaps created by the pandemic. As the HCAs repeatedly performed PPE and hand hygiene rituals accurately, their self-doubt diminished while their self-assurance grew. Mastering these key skills fostered an emerging sense of efficacy that motivated them to meet the challenges posed by COVID-19 patients. Demonstrating mastery experiences built their belief in their capabilities to execute the necessary infection control practices to both take care of contagious patients and protect themselves. Their competence and control in adhering to the guidelines strengthened their self-confidence that they could perform such actions successfully, even under stressful conditions. One HCA reported the following:
*“Now I can suit up in PPE seamlessly. My hands know what to do. The guidelines ingrained those steps until it became automatic. By learning with others, I have not only picked up these important skills but also gained the confidence to use them without hesitation.”*(P05, female)

### 3.3. Theme 3: Fostering Collective Resilience through Shared Support

The participants emphasized the importance of supportive connections in helping them overcome the challenges posed by COVID-19. This supportive framework offered guidance, constructive feedback, and emotional support from nursing staff, physicians, coworkers, and family members.

#### 3.3.1. Team Support

Mentorship from experienced nurses shaped the participants’ understanding of their role within the larger healthcare team. Their daily collaborative efforts fostered a sense of belonging and appreciation. All participants emphasized a collective spirit of mutual encouragement and partnership across the ward staff. One exemplary quote from an HCA is as follows:
*“We supported each other, regardless of positions—doctors, nurses, assistants. The nurses exuded confidence and encouraged me when I was scared. This camaraderie has nurtured my passion for caring for COVID-19 patients.”*(P16, female)

The spiritual guidance from nurses and the gratitude from the public also buoyed participants. Frequent caring words from coworkers provided essential emotional support. This culture of teamwork and solidarity empowered them to overcome shared challenges. The participants credited this supportive community with helping build their resilience and readiness to deliver attentive care during an unprecedented crisis. One HCA described the following:
*“My coworkers constantly uplifted my spirits. We were a family working together, which streamlined the process for all of us involved.”*(P25, male)

#### 3.3.2. Building Collective Capacity through Collaborative Learning

The participants emphasized how mutual support provided opportunities for collaborative learning and growth during the pandemic. They felt comfortable approaching nurses for guidance when unsure, facilitating engagement across roles. Through this collaborative educational experience, the HCAs were able to expand their capabilities and deepen their connections with the nursing staff. The nurses shared insights and modeling techniques that allowed the assistants to improve their competencies. This type of learning nurtured a collective capacity across the team to deliver attentive care amidst unprecedented challenges. One supportive quote from an HCA is as follows:
*“I am reminded of working during the 2003 SARS outbreak in Hong Kong. Caring for SARS patients with limited virus knowledge was extremely stressful. The nurses I worked with provided invaluable mentorship, teaching me infection control and safe patient care. I learned so much from their expertise and compassion during that difficult period. My caregiver skills grew exponentially thanks to their support. Now, with COVID-19, I feel better prepared to provided skilled, empathetic care due to my SARS experience. Although challenging, that early experience equipped me to handle outbreaks like COVID-19. I aim to apply the nursing knowledge and values I learned from my earlier mentors. Their example guides me a caregiver during crisis.”*(P20, female)

The participants credited their strengthened skills gained through this team mentoring with empowering them to better support the nurses and take on new responsibilities. Their readiness to meet emerging demands grew through these supportive connections focused on building collective knowledge and resilience. One HCA described the following:
*“My collaboration with nurses substantially improved during the pandemic, especially for procedures. I’d prepare equipment while a nurse offered help if needed. Our teamwork enhanced communication and my readiness for more challenges yet to come in this crisis.”*(P05, female)

#### 3.3.3. Skill Building through Informal Guidance

Some participants indicated that a designated leader for them may not be possible within their work environments where experienced HCAs are not readily available in all shifts. Under the circumstances where formal guidance appeared limited, some HCAs turned to informal guidance (i.e., persons who may not be officially within their reporting line) by occasionally seeking guidance from nurses or HCAs with more working experience in neighboring wards. They eagerly engaged in discussions to obtain insights that would aid their work.
*“My ward does not have a designed leader for me. The senior coworkers [experienced HCAs] and nurses in another ward was invaluable since I lacked prior training on COVID-19. Their informal guidance allowed me to provide competent care. This informal learning experience has empowered me to contribute to the development of a shared understanding and perform care which has resulted in increased efficiency, saved time and effort.”*(P18, female)

The participants emphasized that informal guidance from the nurses within the same hospital helped build their competencies in this unfamiliar context. The participants frequently sought guidance from nurses or coworkers to perform their duties effectively. They eagerly engaged in discussions to obtain insights that would aid their work.
*“My skills improved exponentially thanks to the nurse’ kind guidance and willingness to provide workarounds and answer all my questions.”*(P03, female)

### 3.4. Theme 4: Self-Efficacy as a Catalyst for Adaptive Growth 

This theme delves into the crucial role of self-efficacy in driving the adaptive growth experienced by HCAs during the COVID-19 pandemic. It highlighted how a heightened sense of self-belief, nurtured through collaborative learning and collective resilience, empowered these frontline workers to embrace new challenges and responsibilities, ultimately paving the way for the transformative journeys described in the next theme.

#### 3.4.1. The Empowering Cycle of Self-Efficacy and Adaptative Growth

The pandemic presented HCAs with a steep learning curve, demanding them to rapidly acquire new skills and knowledge to effectively care for COVID-19 patients. This is where self-efficacy—the belief in one’s ability to succeed—played a pivotal role. As participants stepped outside their comfort zones and successfully navigated unfamiliar terrain, their confidence in their own capabilities grew, further fueling their willingness to adapt and embrace new challenges. 

Participants described mixed reactions when first caring for COVID-19 patients. While some viewed it as an opportunity for growth, most felt apprehension about the infection risks.
*“I had to learn new skills quickly to care for COVID-19 patients. My self-belief was key in this process. I pushed myself out of my comfort zone and took on new challenges, which boosted my confidence. I thoroughly cleaned myself and belongings before leaving work each day. I did not want to infect my daughter. Every time I was caring for a patient, I reinforced my belief that I could handle the demands of the pandemic. This increased my confidence encouraged me to take on, learn new things, and adapt to changing protocols. I think it is a cycle of growth that helped me get through the crisis.”*(P19, female)

The situation created a powerful positive feedback loop: each instance of providing compassionate care to a critically ill patient reinforced their belief in their ability to handle the demands of the pandemic. One participant described the following:
*“Prayer and faith provided support during this transition. By leaning on their spiritual beliefs, some of my HCA colleagues discovered reservoirs of strength. To me, their faith and journeys represent powerful feedback. My faith helps me cope with new responsibilities. I trust in a higher power to guide me as I fulfill my role.”*(P17, female)

This heightened self-efficacy, in turn, emboldened them to take on more responsibilities, seek out new learning opportunities, and readily adapt to evolving protocols and procedures. By facing each unfamiliar experience with resolve, they expanded their capabilities to meet any emerging challenges.

#### 3.4.2. Collective Resilience as a Springboard for Growth

The supportive and collaborative learning environment within healthcare teams proved essential in nurturing this increased self-efficacy. The camaraderie and shared experiences among HCAs fostered a sense of collective resilience. They learned from each other, shared coping mechanisms, and celebrated successes together, reinforcing a collective belief in their ability to overcome challenges. The collective resilience fostered within healthcare teams provided a fertile ground for individual adaptive growth. The shared commitment to providing quality care, even amidst a global health crisis, created a powerful sense of purpose and motivation. This collective resilience provided a safety net, allowing individuals to take risks, make mistakes, and learn from their experiences without fear of judgment, ultimately accelerating their adaptive growth.

Enhanced collaboration with the nurses also empowered the assistants to deliver more vigilant care that was attuned to patients’ needs. By closely monitoring the breathing patterns and cues, they gained insight into how to provide attentive support.
*“I listened for subtle changes in patients’ breathing and voices to understand their condition.”*(P08, female)
*“The nurses kept us informed and my openness for change allowed me to respond quickly to any status changes. We shared our experiences among colleagues; we are a team and work together to face new challenges in each shift. This is new for us to take risks and make mistakes without being afraid of any judgment. It was a powerful reminder that I was not alone in this fight, and that together, we could overcome even the most dauting challenges.”*(P07, male)

Through this proactive peer protection and guidance from nurses, participants were able to build purposeful capabilities to safely fulfill their duties during an unpredictable crisis. Their assertive efforts to implement prudent procedures nurtured the collective knowledge, resilience, and readiness required to meet emerging demands.

### 3.5. Theme 5: Paving the Way for Transformation

The interplay of increased self-efficacy, collaborative learning, and collective resilience created a powerful engine for adaptive growth among HCAs. By embracing new challenges, mastering new skills, and drawing strength from their colleagues, they underwent a process of profound professional and personal development. This newfound confidence and resilience laid the foundation for the transformative experiences explored in this perspective, where these frontline workers emerged from the crucible of the COVID-19 pandemic with a renewed sense of purpose, enhanced skills, and an unshakeable belief in their own capabilities. Participants struggled emotionally while caring for critically ill COVID-19 patients with limited resources, with one participant describing the following:
*“It was disheartening to witness each patient supplanted by another embroiled in the same battle, emphasizing the pandemic’s impact.”*(P23, male)

Despite personal risks and inadequate staffing, the assistants courageously maintained compassionate care. Their roles were expanded extensively to satisfy escalating needs.
*“As HCAs, we cannot abandon patients regardless of amplified workloads and hazards. Every team member is indispensable presently.”*(P15, female)

Through this trial by fire, the participants underwent a profound personal and professional transformation. By harnessing their inner tenacity and purpose during hardship, they unearthed latent reserves of fortitude and capabilities.
*“I was oblivious to the magnitude of my resilience until now—this has transformed me beneficially.”*(P09, female)

Hence, confronting quintessential healthcare challenges directly catalyzed their evolution. The participants emerged with greater confidence in their aptitude to manage future crises due to the resilient self-efficacy that they accrued during the pandemic.

## 4. Discussion

This study helps to investigate the firsthand experiences of HCA personnel in Hong Kong, lends critical perspectives that shine a light on the specific realities they face, and enriches our global understanding of readying for future crises by providing an intricate comprehension of their challenges as well as pinpointing vital lessons relevant internationally. The findings of this study delineate the enduring endeavors of HCAs to strengthen their roles in healthcare settings during the COVID-19 pandemic. The strengthened roles could involve HCAs working to their greater potential, with increased involvement in many aspects of patient care and in the collaborative interrelationship between nurses and HCAs [31,35,47,52]. The availability of equipment, protocols, guidelines, and personnel provided frontline reinforcement for the HCAs—facilitating their self-management of intense emotions, enhancing their proficiency in fulfilling their responsibilities, and aiding their transition to new roles when required. In the face of the pandemic, taking on new tasks such as proactive self-education on ever-changing infection control protocols signified an additional workload for the HCAs. During the COVID-19 pandemic, insufficient frontline reinforcement was a frequent grievance throughout hospital systems [13,53,54]. Cognizance of available frontline reinforcements was crucial to effectively managing workforce allocation, particularly in terms of augmenting personnel numbers, guaranteeing the safety of both staff and patients, and improving the operational resilience of healthcare organizations during high-stress situations [53].

The ICN has indicated that during the COVID-19 pandemic, numerous frontline healthcare workers [40], including HCAs, experienced disparities in the safety measures and supplementary assistance they were granted compared to physicians and nurses. Examples of such discrepancies included the provision of accommodation, financial incentives, and sustenance [55,56,57]. This situation raised concerns regarding the equitable distribution of support and protective measures among the various frontline healthcare providers working to combat the pandemic. In Western countries and Hong Kong, social media highlighted the contributions of physicians and nurses and brought attention to the need for targeted interventions and support measures to address their mental health burden [56,57,58,59,60]; however, not many contributions by HCAs were mentioned in the literature. It is possible that the insufficient recognition of contributions by the general public may have influenced the allocation of essential assistance and resources, potentially causing disparities among various groups within the healthcare workforce. This situation underscores the need to raise awareness regarding the valuable contributions of all healthcare providers and to ensure an equitable distribution of support and resources throughout the sector.

Despite this, our findings reveal that HCAs frequently engaged in conversations within their work environment about access to frontline reinforcement. They remarked that receiving continual and substantial support from nurses fostered a sense of respect, motivating them to contribute further. This dynamic support fostered a symbiotic relationship between HCAs and nurses, characterized by mutual assistance and collaboration. This strong bond was established through ongoing, in-depth interactions encompassing open dialogues, attentive listening, and collaborative efforts, all of which are essential to the development of resilience among HCAs. The reinforcement and cooperation between HCAs and nurses played a crucial role in the effective functioning of the healthcare system, ultimately benefiting patient care during the pandemic [61,62]. Additionally, our findings also suggest that for resilience among HCAs to develop, open communication and collaborative learning between healthcare providers are necessary. Under these conditions, HCAs are able to obtain existential support. Through the respect fostered between healthcare providers, HCAs can appreciate the valuable roles played by all team members.

Furthermore, the findings of this study highlighted valuable opportunities on which healthcare institutions should capitalize. Those in management positions should acknowledge and demonstrate an appreciation for HCAs by offering development opportunities, motivation, and rewards. Healthcare organizations can foster a positive work environment by addressing these aspects, ensuring that HCAs feel valued and respected for their indispensable contributions to patient care. This, in turn, may facilitate the development of resilience among HCAs, ultimately benefiting the institution and the quality of care provided. In previous studies, healthcare workers have reported feelings of fatigue, pressure, and anxiety [55,63,64]. Management should ensure that all staff members receive appropriate assistance and resources to cope with the demands and challenges of their roles, ultimately fostering a healthier and more supportive work environment. In our study, HCAs often expressed their emotions by engaging in contemplative practices and discussing their faith with others. Colleagues served as essential sources of spiritual encouragement, assisting them to cope with the difficulties presented by the pandemic.

Concerning the availability or insufficient knowledge, the details of pandemic-related knowledge provided to the HCAs affected their sense of readiness, fear, and anxiety. As such knowledge is relevant to HCAs’ work, the insufficiency also seemed to influence the HCAs’ perception of their competence. These perceptions and emotional responses paralleled those expressed during other unforeseen crises, such as the SARS and MERS pandemics [65,66]. In this study, participants reported acquiring healthcare information by exchanging knowledge during training sessions within their work environments. With an increased pandemic-related knowledge level, HCAs may experience less anxiety and/or pessimistic outlooks, which results in enhanced care quality and elevated job satisfaction during the COVID-19 pandemic [67,68]. Based on our study findings, HCAs’ experiences during the COVID-19 pandemic echo those from the SARS and MERS outbreaks, with shared elements of amplified workload and emotional stress [12,69]. Healthcare institutions can address this stress by providing mental health services, encouraging open communication, and maintaining manageable workloads [70]. Nonetheless, the global span and endurance of the COVID-19 pandemic posed distinctive challenges. A comprehensive comparative study is needed to guide the development of effective strategies to assist HCAs in future pandemics.

In an era when learning can happen through various virtual means [71,72], our study recognizes the potential of promoting skill-enhancement training through pamphlets or digital platforms. However, our findings also highlight the importance of preserving hands-on training, particularly during situations such as the pandemic. The participants expressed a clear preference for acquiring new knowledge through in-person demonstrations, underscoring the value they place on non-digital, face-to-face interactions with healthcare professionals. Their accounts emphasize the unique insights gained through this traditional approach.

This study offers important insights into the impact of self-efficacy on HCAs’ psychological well-being during the COVID-19 pandemic in Hong Kong. These findings, while specific to the cultural and systematic contexts of Hong Kong, may inform strategies to promote self-efficacy and morale among HCAs in healthcare settings worldwide, thereby enhancing their welfare and performance [73,74]. Based on our findings, self-efficacy develops gradually among HCAs as they continually make adaptive adjustments in the face of the pandemic. Our findings support that collaborative learning may play a role in increasing the self-efficacy among HCAs in managing the new difficulties and challenges in their work. Beyond self-efficacy, our findings align with previous literature suggesting that collaborative learning (which involves learning through working together with peers and other healthcare professionals) was crucial in the development of resilience among participants [19,20]. Collaborative learning involved exchanging information, sharing knowledge, and providing mutual support among stakeholders [19,20]. Through coordinated efforts, negotiation, and aligned actions, these stakeholders ensured safe care delivery. With adaptive adjustments and self-growth in the face of realities, HCAs gradually developed resilience. Resilience refers to a system’s ability to reduce susceptibility to adverse conditions and adapt to novel situations [19,20]. In our study, participants operated in an environment addressing the fundamental requirements of COVID-19 patients. This adaptability optimizes the core functions of HCAs within their work environments [19,20].

HCAs are integral members of the healthcare workforce who provide crucial frontline care, as seen during public health crises such as the COVID-19 pandemic [75]. However, limited research exists that examines the firsthand perspectives and experiences of HCAs themselves regarding the development of self-efficacy and resilience during the care provision during COVID-19 [58]. This study addressed the above-mentioned knowledge gap using a thematic analysis approach. In fact, Hong Kong has previous experience responding to infectious outbreaks, such as the 2003 SARS pandemic [76]. Recommendations were put forth after SARS to better empower frontline healthcare workers, including ensuring adequate supplies and ongoing training related to bedside skills. More than the hardware support previously mentioned, the participants in this study demonstrated that resilience is important in addressing public health crises and that resilience among HCAs builds on self-efficacy, self-growth, collaborative learning, and support from various sources including professional clinicians and working organizations. More qualitative research from interpretive traditions is needed to fully understand the meanings of the contextual factors that may hinder or promote the development of resilience among this important group of healthcare workers.

### 4.1. Strengths and Limitations

The findings from this study imply that the participants were potentially able to overcome certain workplace tensions and stress. There is minimal evidence-based research from relevant clinical settings in different locations with which the study can be compared. Furthermore, these issues have not yet been explored using this approach in Hong Kong. Thus, to address this limitation, future studies can include a more diverse sample that encompasses HCAs with varying social and familial support levels. This approach will deliver a more expansive understanding of the situations encountered by HCAs throughout the COVID-19 global health crisis. Nonetheless, our findings offer useful initial insights into ways to strengthen HCA capabilities. The data can inform targeted enhancements to training programs, resource allocation, workplace policies, and practices to further build the resilience of this essential group of healthcare workers for future public health outbreaks.

### 4.2. Implications for Healthcare Practice and Policy

This study emphasizes the pivotal roles of HCAs within the healthcare domain, especially during the COVID-19 pandemic. In response to the pandemic, healthcare systems worldwide were confronted with preparedness challenges. The findings of this study imply that the resilience developed among HCAs is instrumental in building a functional and adaptive workforce that supports clinical services in a pandemic. This study identified HCA enablement as an instrumental factor in enhancing readiness, advocating for its incorporation into disaster management protocols. This research encourages deliberate efforts to reinforce HCA self-efficacy within healthcare systems. Based on the findings, recommendations for further action encompass the following:-Conducting additional research to explore the effects of varied organizational structures on the development of resilience among HCAs via comparative or longitudinal approaches;-Fostering more supportive work environments to optimize HCA job satisfaction and self-efficacy;-Implementing training programs to nurture HCAs with competencies for caring for patients in unpredictable situations (in this case, the COVID-19 pandemic).

## 5. Conclusions

This study focused on understanding and addressing HCAs’ individual perceptions within their work environments during public health crises. The self-efficacy of HCAs can contribute to the preparedness of healthcare systems during crises, an essential aspect of disaster readiness planning. In coping with the COVID-19 pandemic, HCA self-efficacy and resilience are crucial factors for successful adaptation. As healthcare organizations navigate the pandemic and potential future public health crises, strategies to empower HCAs should be an integral element of organizational transformation processes. Consequently, this study advocates for a strategic emphasis on enhancing HCA self-efficacy and resilience within healthcare systems.

## Figures and Tables

**Table 1 healthcare-12-01544-t001:** Interview guide.

No.	Probing Questions
1.	Could you describe the main psychological feelings you have experienced while caring for patients in a clinical setting during the COVID-19 pandemic?
2.	What are your coping strategies?
3.	What insights have you gained from facing the COVID-19 pandemic?
4.	How did you feel when accepting the anti-pandemic task?
5.	What were your emotions when providing care for patients amidst the COVID-19 pandemic?
6.	Are there any changes in your life that have led you to adopt new learning strategies or adaptive forms of coping?
7.	What methods do you use to deal with changes in your work duties and personal life?
8.	How do you perceive and feel about this task related to preventing the spread of the pandemic?

**Table 2 healthcare-12-01544-t002:** Participant characteristics.

Characteristics		*N* = 25	
		*n*	%
Age (years)			
	30–34	2	8
	35–40	5	20
	41–46	18	72
Gender			
	Male	4	16
	Female	21	84
Education level			
	Secondary	20	80
	Tertiary	2	8
	Higher diploma	3	12
Marriage and offspring			
	Unmarried without children	2	8
	Married without children	1	4
	Married with children	22	88
No. of year(s) in HCA post			
	1–5	2	8
	6–10	6	24
	11–15	10	40
	16–20	7	28
No. of month(s) in current post			
	3–6	14	56
	7–10	10	40
	>10	1	4
Working unit/Department			
	Medical/Surgical	21	84
	Accident and Emergency	2	8
	Pediatric	2	8
Religious belief			
	None	6	24
	Christian	12	48
	Catholic	5	20
	Buddhist	2	8

**Table 3 healthcare-12-01544-t003:** Themes and sub-themes of this study.

Themes	Sub-Themes
Frontline Reinforcement: Supporting HCAs through Resourcing and Education Amidst the COVID-19 Crisis	- Heightened Apprehension Surrounding Perceived Infection Risks
	- Proactive Self-Education on Infection Prevention Protocols
	- Challenges Adjusting to Evolving Practices
Confronting Uncertainty: Building Personal Fortitude in the Face of the COVID-19 Pandemic	- Fostering Routines through Repetition
	- Building Self-Confidence through Mastering Skills
Fostering Collective Resilience through Shared Support	- Team support
	- Building Collective Capacity through Collaborative Learning
	- Skill Building through Informal Guidance
Self-Efficacy as a Catalyst for Adaptive Growth	- The Empowering Cycle of Self-Efficacy and Adaptative Growth
	- Collective Resilience as a Springboard for Growth
Paving the Way for Transformation	- Paving the Way for Transformation

## Data Availability

Data sharing is not applicable to this article due to the small number of participants involved. All participants are currently employed in the positions described in the study. The study’s design priorities the protection of participant identities, therefore, individual data will not be shared.

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
