# Peer review of "Navigating Uncertainty with Compassion: Healthcare Assistants’ Reflections on Balancing COVID-19 and Routine Care through Adversity"

_healthcare, 2024, doi:10.3390/healthcare12151544_

Round 1

Reviewer 1 Report (New Reviewer)

Comments and Suggestions for Authors

Dear Editor,

Thank you for the opportunity to review this manuscript. Below, please find some comments and suggestions for the authors:

  1. Did any participant withdraw during data collection? Please mention it in the manuscript.
  2. Were there any repeated interviews? If yes, what was the reason?
  3. Methods: How to maintain the trustworthiness of this qualitative study? Please add a sub-section about the rigor and trustworthiness of the study under the methods section.
  4. Result: How many participants were female and how many were male? Please add the characteristics of gender to Table 1. This is related to the results, especially when authors write the description of participants with a P number and gender after each excerpt.
  5. It would be beneficial if all themes and sub-themes are presented in the table, figure, or diagram to make it easier for readers to see the overall themes.
  6. On page 3, line 113, “Participants voluntary participant in this investigation provided their written informed consent prior to study engagement.” What does it mean by participants voluntary participant? Please revise the sentence.

Author Response

Response to Reviewer 1 Comments

Dear Respected Reviewer,

Healthcare

RE: Manuscript titled “Navigating uncertainty with compassion: healthcare assistants’ reflections on balancing COVID-19 and routine care through adversity”

Thank you for your unstinting effort to review the revised manuscript. Our research team appreciated your valuable comments. In this document, we provided our responses in a point-by-point format. We hope that our revisions to the manuscript can address your concerns satisfactorily.

This response letter was prepared with reference to a template that we downloaded from the Journal website. In that template, we noticed that all responses from the authors were highlighted in red. We thus followed the prescribed style and format.

Yours faithfully,

Manuscript no.: healthcare-3078683

Point 1: Did any participant withdraw during data collection? Please mention it in the manuscript.

Response 1: Thank you for your comment. No participant withdraws during data collection, our research team revised as follows:

“Therefore, data saturation was attained during the 23rd interview, after which no new

insights emerged within the final two interviews, and no participant withdrew during

data collection, indicating cessation of the data collection.” [Line 156-158]

Point 2: Were there any repeated interviews? If yes, what was the reason?

Response 2: No repeat interviews were conducted in this study, ensuring that each participant provided their unique perspectives and experiences during a single interview session.

Point 3: Methods: How to maintain the trustworthiness of this qualitative study? Please add a sub-section about the rigor and trustworthiness of the study under the method section.

Response 3: Thank you for your advice. Our research team add a sub-section about the rigor and trustworthiness of the study in “2.5. Rigor and trustworthiness” section in Line 174-181.

Point 4:  On page 3, line 113, “Participants voluntary participant in this investigation provided their written informed consent prior to study engagement.” What does it mean by participants voluntarily participant? Please revise the sentence.

Response 4: Thank you for your comment. We have carefully review as follows:

“Participation was entirely voluntary, participant retained the right to withdraw with-out penalty, and informed consent was obtained after a comprehensive explanation of the study’s purpose, procedure, and potential risks and benefits.” [Line 123-125]

Point 5: Are the methods adequately described? (x) Can be improved

Response 5: Agree. We have carefully reviewed your comments and agree that the Methods section can be improved. Our team has revised address the concerns raised in Point 1, 2, 3, and 4, providing clearer and more detailed information about our study methodology. We believe these changes significantly enhance the clarity and comprehensiveness of the Methods section. [Line 108-181]

Point 6: Result: How many participants were female and how many were male? Please add the characteristics of gender to Table 1. This is related to the results, especially when authors write the description of participants with a P number and gender after each excerpt.

Response 6: Thank you for your comment. A total of 25 individuals, consisting of 4 male and 21 female participants, were included in this study and added the characteristics of gender to Table 1. [Line 191-192]

Point 7:  It would be beneficial if all themes and sub-themes are presented in the table, figure, or diagram to make it easier for readers to see the overall themes.

Response 7: Agree. Our research team add the following table in the “3. Results” section in Line 193 as follows:

      Table 3. Themes and sub-themes of this study.

Themes

Sub-themes

Frontline Reinforcement: Supporting HCAs through Resourcing and Education Amidst the COVID-19 Crisis

-            Heightened Apprehension Surrounding Perceived Infection Risks

-            Proactive Self-Education on Infection Prevention Protocols

-            Challenges Adjusting to Evolving Practices

Confronting Uncertainty: Building Personal Fortitude in the Face of the COVID-19 Pandemic

-            Fostering Routines through Repetition

-            Building Self-Confidence through Mastering Skills

Fostering Collective Resilience through Shared Support

-            Team support

-            Building Collective Capacity through Collaborative Learning

-            Skill Building through Informal Guidance

Self-Efficacy as a Catalyst for Adaptive Growth

-            The Empowering Cycle of Self-Efficacy and Adaptative Growth

-            Collective Resilience as a Springboard for Growth

Paving the Way for Transformation

-            Paving the Way for Transformation

Point 8: Are the results clearly presented? (x) Can be improved

Response 8: Thank you for your advice. We have carefully reviewed your comments and agree that the Methods section can be improved. Our team has revised address the concerns raised in Point 6 and 7, providing clearer and more detailed information about our study results. We believe these changes significantly enhance the clarity and comprehensiveness of the Results section. [Line 183-193]

Reviewer 2 Report (New Reviewer)

Comments and Suggestions for Authors

* Does the introduction provide sufficient background and include all relevant references? 

I was particularly impressed with the statistics concerning the number of infected individuals and the global deaths resulting from COVID-19. They are sobering facts that need more attention from the general audience. Now that we are in a long-term COVID status, not much is publicized concerning what to look for in symptoms, how home care may help, and when to seek further definitive care.

* Is the research design appropriate?

The design was sufficiently described, including the guidelines as a framework for the study (COREG).

* Are the methods adequately described?

This manuscript is well put together, with the correct elements of the study documented (qualitative method, data collection, ethics, analysis, etc.). The presentation would be easily replicated in another study.

* Are the results clearly presented?

The results and participant quotes substantiate the identified themes of this study and are articulated for ease of comprehension.  

* Are the conclusions supported by the results?                                                               

The conclusions substantiate and support the results and discussion of this manuscript. Limitations & implications are included for further thought.

Comments on the Quality of English Language

This document represents a thorough understanding of the grammar and syntax involved in moving from a language you are comfortable with to another that may be uncomfortable (foreign). You have all presented an easy reading, and I commend you. 

Author Response

Response to Reviewer 2 Comments

Dear Respected Reviewer,

Healthcare

RE: Manuscript titled “Navigating uncertainty with compassion: healthcare assistants’ reflections on balancing COVID-19 and routine care through adversity”

Thank you for your unstinting effort to review the revised manuscript. Our research team appreciated your valuable comments. In this document, we provided our responses in a point-by-point format. We hope that our revisions to the manuscript can address your concerns satisfactorily.

This response letter was prepared with reference to a template that we downloaded from the Journal website. In that template, we noticed that all responses from the authors were highlighted in red. We thus followed the prescribed style and format.

Yours faithfully,

Manuscript no.: healthcare-3078683

Point 1: Minor editing of English language required. (x)

Responses 1: We agree with this suggestion. Our research team has consulted a senior editor with over 10 years of academic editing experience to address the minor English language issues in our revised manuscript. The editor’s expertise and meticulous review have significantly improved the language, grammar, and overall readability of our work, ensuring that the revised version meets the high standards expected by ‘Healthcare’.

Point 2: The presentation would be easily replicated in another study.

Response 2: Thank you for bringing to our attention the concerns regarding the replicability of the ‘Methods’ section in our manuscript due to the unique circumstances of the COVID-19 pandemic. We appreciate your feedback and the opportunity to clarify our approach.

We acknowledge that the COVID-19 pandemic has created an unprecedented public health crisis, which has posed challenges for conducting research and collecting data. Our study was designed as a special initiative to address the pressing need for timely and relevant research during this critical period.

In the ‘Methods’ section of our manuscript, we have aimed to provide a detail and transparent account of our research methodology, data collection procedures, ethical considerations, and data analysis techniques. While the specific circumstances of the COVID-19 pandemic may not be easily replicable in future studies, we believe that the rigorous and systematic approach we have employed can serve as a valuable reference for researchers conducing similar studies in the context of public health crises.

Our team have made every effort to ensure that our methods are clearly described well-justified, allowing other researchers to understand and potentially adapt our approach to their own research contexts. The unique nature of the COVID-19 pandemic has necessitated innovative and flexible research methods, and we believe that sharing our experience and insights can contribute to the collective knowledge and best practices in conducting research during such challenging times. [Line 108-181]

Reviewer 3 Report (New Reviewer)

Comments and Suggestions for Authors

The article highlights the importance of the work of a category of health professionals who are easily forgotten, so congratulations on the choice of the population indicating their need for services.

The article is well organised and structured, with no suggestions for improvement. It really needs to be published so that these professionals are better recognised.

Author Response

Dear Respected Reviewer,

Healthcare

RE: Manuscript titled “Navigating uncertainty with compassion: healthcare assistants’ reflections on balancing COVID-19 and routine care through adversity”

Thank you for your unstinting effort to review the revised manuscript. Our research team appreciated your valuable comments. I am thrilled to learn that my manuscript has been accepted the work of our team. It is an honour to have my work recognized by such a prestigious journal, and I am grateful for the opportunity to share my findings with the scientific community.

Once again, I would like to express my heartfelt appreciation to the editorial team and the reviewers for their diligence, professionalism, and support throughout the review process. Your commitment to advancing scientific knowledge and facilitating the dissemination of high-quality research is truly admirable.

Thank you for considering my manuscript and for the opportunity to contribute to Healthcare. I look forward to future collaborations and to continue engaging with the exceptional research community that your journal fosters.

Yours faithfully,

Manuscript no.: healthcare-3078683

Reviewer 4 Report (New Reviewer)

Comments and Suggestions for Authors

1. In Author Affiliation it is stated: "1School of Health Sciences, Saint Francis University, 2 Chui Ling Lane, Tseung Kwan O, New Territories, Hong Kong, China". What does the word 2 mean before Chui Ling Lane? Affiliation 2: "Hong Kong Institute of Paramedicine, Hong Kong, China"? Please examine carefully the correct way to write affiliation according to the journal guidelines.

2. Introduction: In the Introduction, the term healthcare assistants is abbreviated as "HCAs". However, the reviewer found many inconsistencies in the writing of the abbreviation, sometimes writing "HCAs", sometimes writing "HCA" sometimes writing "HCA's". Are these three abbreviations different things? or just typos?

3. In the introduction, the study's importance has been clearly stated; however, it fails to specify the differences between this study and previous ones. What sets this study apart?

4. Methods: In Lines 130-131, I think the hyphenation of the word "vide-oconferencing" is not quite right. If it can be edited, write "video-conferencing" which I think is easier to read.

5. Participants: In lines 118-120, the number of participants is stated as 25 people. Where did that number come from? How many people are in the original population? Is there a specific formula for determining the number of participants?

6. Data: The author used a questionnaire in their study, which was validated by experts. However, it has not been stated what theory, regulation, or standard was used as the basis for compiling the questions.

7. Table 2. Participant characteristics. Please type and arrange the table properly.

8. The "4. Discussion" can be bolded on line 452.

9. The author has cited an excessive number of sources for a single sentence, which is unnecessary. Please review references 17-25 (lines 64-67), 17-27 (lines 67-68), and 52-60 (lines 478-481).

Author Response

Response to Reviewer 4 Comments

Dear Respected Reviewer,

Healthcare

RE: Manuscript titled “Navigating uncertainty with compassion: healthcare assistants’ reflections on balancing COVID-19 and routine care through adversity”

Thank you for your unstinting effort to review the revised manuscript. Our research team appreciated your valuable comments. In this document, we provided our responses in a point-by-point format. We hope that our revisions to the manuscript can address your concerns satisfactorily.

This response letter was prepared with reference to a template that we downloaded from the Journal website. In that template, we noticed that all responses from the authors were highlighted in red. We thus followed the prescribed style and format.

Yours faithfully,

Manuscript no.: healthcare-3078683

Point 1: Does the introduction provide sufficient background and include all relevant references? (x) Must be improved

Response 1: Thank you for your comment. Following the revision process, our research team revised the following content in the ‘introduction’ part as follow:

“However, data on infections and deaths among HCAs is severely limited across most nations, highlighting a significant gap in understanding the full extent of their experiences and challenges. The ICN advocated standardizing data collection on cases and fatalities among HCAs, which is crucial to understanding and mitigating the occupational risks they face [40]. While some studies have investigated the impact of the pan-demic on healthcare workers in general, limited qualitative research has specifically explored the unique perspectives and experiences of HCAs themselves, particularly during times of crisis, like those seen during the COVID-19 pandemic [41]. This study aims to address this gap by focusing exclusively on the voices of HCAs, providing valuable insights into their resilience and self-efficacy during this unprecedented time. The findings of this study will provide a greater understanding of HCA perspectives, ad-dressing gaps in knowledge about the development of resilience and self-efficacy of HCAs in both within the localized context of Hong Kong and the broader international landscape. By delving into the specific challenges, coping mechanisms, and support systems of HCAs in Hong Kong, this study offers a unique contribution to the existing literature, which has largely overlooked this critical workforce.

The study aimed to explore HCAs’ experiences and perspectives on resilience and self-efficacy during the COVID-19 pandemic in Hong Kong. Based on the findings of this study, management of the healthcare system can identify ways to better support and empower this critical workforce during future public health crises. By under-standing the specific needs and challenges faced by HCAs, healthcare organizations can develop targeted interventions and policies to enhance their resilience and self-efficacy, ultimately strengthening the healthcare system’s response to future

pandemics or other crises.” [Line 84-106]

Point 2: To examine carefully the correct way to write affiliation according to the journal guidelines.

Response 2: Agree. Our research team provided correct way to write affiliation according to the guidelines:

“Alice YIP 1,*, Jeff YIP 2, Zoe TSUI 1, and Graeme Drummond SMITH 1

1 S.K. Yee School of Health Sciences, Saint Francis University, Hong Kong, China;

  [email protected] (Z.T);

2 Hong Kong Institute of Paramedicine, Hong Kong, China; [email protected]

  (J.Y.)” [Line 5-8]

Point 3: Introduction: in the introduction, the term healthcare assistants is abbreviated as “HCAs”. However, the reviewer found many inconsistencies in the writing of the abbreviation, sometimes writing “HCAs”, sometimes writing “HCA” sometimes writing “HCA’s”. Are these three abbreviations different things? Or just typos?

Response 3: Thank you for your advice. In Line 14, this is typo error ‘healthcare assistant (HCA)’ instead of healthcare assistants (HCAs). The term ‘HCAs’ is an abbreviation for ‘healthcare assistants’. When written this way, ‘HCAs’ is referring to healthcare assistants in the plural form. For example:

“However, studies suggest HCAs were frequently deputized, their skills not fully lever-

aged, and their contributions overlooked during the pandemic.” [Line 46-47]

In this case, ‘HCAs’ means multiple healthcare assistants.

The phrase ‘healthcare assistants’ is the full, unabbreviated term. It refers to medical support staff who work under the supervision of nurses or other healthcare professionals to provide basic patient care. For example:

“Healthcare assistants are vital members of the healthcare team, as their work has been

shown to improve patient comfort during hospitalization.” [Line 49-50]

Finally, the term ‘HCA’s’ within an apostrophe before the ‘s’ is a way of showing possession, as in something belonging to a healthcare assistant. However, this is not the correct usage when simply referring to healthcare assistants in the plural. For example:

“Self-efficacy, defined as a HCA’s belief in their own capabilities to perform actions and achieve outcomes, has been thought to have direct bearing on individual performance in particular contexts.” [Line 58-60]

            In summary:

            HCAs = abbreviation for multiple healthcare assistants

            Healthcare assistants = full term for the job role

            HCA’s = possessive form, not for simple plural

Point 4: Introduction, the study’s importance has been clearly stated; however, if fails to specify the differences between this study and previous ones. What sets this study apart?

Response 4: Thank you for your comment. Following the revision process, our research team revised the following content in the ‘introduction’ part to present the differences between this study and previous ones as follow:

“However, data on infections and deaths among HCAs is severely limited across most nations, highlighting a significant gap in understanding the full extent of their experiences and challenges. The ICN advocated standardizing data collection on cases and fatalities among HCAs, which is crucial to understanding and mitigating the occupational risks they face [40]. While some studies have investigated the impact of the pan-demic on healthcare workers in general, limited qualitative research has specifically explored the unique perspectives and experiences of HCAs themselves, particularly during times of crisis, like those seen during the COVID-19 pandemic [41]. This study aims to address this gap by focusing exclusively on the voices of HCAs, providing valuable insights into their resilience and self-efficacy during this unprecedented time. The findings of this study will provide a greater understanding of HCA perspectives, ad-dressing gaps in knowledge about the development of resilience and self-efficacy of HCAs in both within the localized context of Hong Kong and the broader international landscape. By delving into the specific challenges, coping mechanisms, and support systems of HCAs in Hong Kong, this study offers a unique contribution to the existing literature, which has largely overlooked this critical workforce.

The study aimed to explore HCAs’ experiences and perspectives on resilience and self-efficacy during the COVID-19 pandemic in Hong Kong. Based on the findings of this study, management of the healthcare system can identify ways to better support and empower this critical workforce during future public health crises. By under-standing the specific needs and challenges faced by HCAs, healthcare organizations can develop targeted interventions and policies to enhance their resilience and self-efficacy, ultimately strengthening the healthcare system’s response to future

pandemics or other crises.” [Line 84-106]

Point 5: Methods: In Lines 130-131. I think the hyphenation of the word “vide-oconferencing” is not quite right. If it can be edited, write “video-conferencing” which I think is easier to read.

Response 5: Thank you for your comment. Our research team revised the typo error as follows:

“All interviews were conducted in Chinese by the second author over Zoom video-

conferencing within a private, password-protected virtual space from March 2021 to

December 2021.” [Line 140-142]

Point 6:  Participants in Lines 118-120, the number of participants is stated as 25 people. Where did that number come from? How many people are in the original population? Is there a specific formula for determining the number of participants?

Response 6: Our study included 25 healthcare assistants actively caring for COVID-19 patients in Hong Kong public hospitals. This sample size is appropriate for our qualitative research methodology, which prioritizes depth of information over participant quantity. Participants were recruited through nursing personnel’s social networks, colleagues, and hospital managers. We determined the number of participants based on the principle of data saturation, a standard practice in qualitative research where interviews continue until no new themes emerge. This approach allowed us to gather rich, detailed data sufficient for comprehensive analysis and meaningful insights into the experiences of healthcare assistants during the COVID-19 pandemic. The sample size of 25 aligns with typical guidelines for qualitative studies using in-depth interviews and provide adequate to address our research questions thoroughly. [Line 130-137]

Point 7:  Data: The author used a questionnaire in their study, which was validated by experts. However, it has not been stated what theory, regulation, or standard was used as the basis for compiling the questions.

Response 7: Thank you for your comment. The study employed a qualitative approach using a validated interview question guide. An expert panel, consisting of a qualitative research scholar and a psychological consulting specialist, validated the interview questions to ensure alignment with the study aim and focus. The questions explored participants’ psychological experiences and emotional responses while caring for COVID-19 patients. Topics covered included changes in work duties and personal lives, coping strategies, insights gained, and perspectives on their role in preventing COVID-19 spread. This approach, while not based on a specific theory or regulation, was designed to capture rich, in-depth data about the healthcare assistants’ experiences during the pandemic. [Line 147-154]

Point 8: Are the methods adequately described? (x) Can be improved

Response 8: Thank you for your advice. We have carefully reviewed your comments and agree that the Methods section can be improved. Our team has revised address the concerns raised in Point 5, 6, and 7, providing clearer and more detailed information about our study methodology. We believe these changes significantly enhance the clarity and comprehensiveness of the Methods section. [Line 108-181]

Point 9:  Table 2. Participant characteristics. Pleas type and arrange the table properly.

Response 9: We appreciate your comment regarding ‘Table 2. Participant characteristics’. We assure you that the table’s format will be properly revised and arranged by our editorial colleagues to meet the journal’s formatting requirements and ensure clear presentation of the participant characteristics. [Line 191-192]

Point 10:  The “4 Discussion” can be bolded on line 452.

Response 10: Agree. Our team revised the ‘4. Discussion’ in Line 469.

Point 11:  The author has cited an excessive number of sources for a single sentence, which is unnecessary. Please review references 17-25 (lines 64-67), 17-27 (line 67-68), and 52-60 (lines 478-481).

Response 11: Thank you for your comment. We revised the references as below:

            “This potential lack of mobility and perpetual perception of being ‘bottom of the totem

pole’ can cause HCAs to experience a lack of purpose, loss of motivation, and

emotional exhaustion—all key factors contributing to burnout [17,23–25]. However,

over the past two decades, HCA roles have expanded and gained greater value

[18,26,27].” [line 64-68]

“In Western countries and Hong Kong, social media highlighted the contributions of

physicians and nurses and brought attention to the need for targeted interventions and

support measures to address their mental health burden [56–60]” [Line 495-498].

Response to Reviewer 4 Comments

Dear Respected Reviewer,

Healthcare

RE: Manuscript titled “Navigating uncertainty with compassion: healthcare assistants’ reflections on balancing COVID-19 and routine care through adversity”

Thank you for your unstinting effort to review the revised manuscript. Our research team appreciated your valuable comments. In this document, we provided our responses in a point-by-point format. We hope that our revisions to the manuscript can address your concerns satisfactorily.

This response letter was prepared with reference to a template that we downloaded from the Journal website. In that template, we noticed that all responses from the authors were highlighted in red. We thus followed the prescribed style and format.

Yours faithfully,

Manuscript no.: healthcare-3078683

Point 1: Does the introduction provide sufficient background and include all relevant references? (x) Must be improved

Response 1: Thank you for your comment. Following the revision process, our research team revised the following content in the ‘introduction’ part as follow:

“However, data on infections and deaths among HCAs is severely limited across most nations, highlighting a significant gap in understanding the full extent of their experiences and challenges. The ICN advocated standardizing data collection on cases and fatalities among HCAs, which is crucial to understanding and mitigating the occupational risks they face [40]. While some studies have investigated the impact of the pan-demic on healthcare workers in general, limited qualitative research has specifically explored the unique perspectives and experiences of HCAs themselves, particularly during times of crisis, like those seen during the COVID-19 pandemic [41]. This study aims to address this gap by focusing exclusively on the voices of HCAs, providing valuable insights into their resilience and self-efficacy during this unprecedented time. The findings of this study will provide a greater understanding of HCA perspectives, ad-dressing gaps in knowledge about the development of resilience and self-efficacy of HCAs in both within the localized context of Hong Kong and the broader international landscape. By delving into the specific challenges, coping mechanisms, and support systems of HCAs in Hong Kong, this study offers a unique contribution to the existing literature, which has largely overlooked this critical workforce.

The study aimed to explore HCAs’ experiences and perspectives on resilience and self-efficacy during the COVID-19 pandemic in Hong Kong. Based on the findings of this study, management of the healthcare system can identify ways to better support and empower this critical workforce during future public health crises. By under-standing the specific needs and challenges faced by HCAs, healthcare organizations can develop targeted interventions and policies to enhance their resilience and self-efficacy, ultimately strengthening the healthcare system’s response to future

pandemics or other crises.” [Line 84-106]

Point 2: To examine carefully the correct way to write affiliation according to the journal guidelines.

Response 2: Agree. Our research team provided correct way to write affiliation according to the guidelines:

“Alice YIP 1,*, Jeff YIP 2, Zoe TSUI 1, and Graeme Drummond SMITH 1

1 S.K. Yee School of Health Sciences, Saint Francis University, Hong Kong, China;

  [email protected] (Z.T);

2 Hong Kong Institute of Paramedicine, Hong Kong, China; [email protected]

  (J.Y.)” [Line 5-8]

Point 3: Introduction: in the introduction, the term healthcare assistants is abbreviated as “HCAs”. However, the reviewer found many inconsistencies in the writing of the abbreviation, sometimes writing “HCAs”, sometimes writing “HCA” sometimes writing “HCA’s”. Are these three abbreviations different things? Or just typos?

Response 3: Thank you for your advice. In Line 14, this is typo error ‘healthcare assistant (HCA)’ instead of healthcare assistants (HCAs). The term ‘HCAs’ is an abbreviation for ‘healthcare assistants’. When written this way, ‘HCAs’ is referring to healthcare assistants in the plural form. For example:

“However, studies suggest HCAs were frequently deputized, their skills not fully lever-

aged, and their contributions overlooked during the pandemic.” [Line 46-47]

In this case, ‘HCAs’ means multiple healthcare assistants.

The phrase ‘healthcare assistants’ is the full, unabbreviated term. It refers to medical support staff who work under the supervision of nurses or other healthcare professionals to provide basic patient care. For example:

“Healthcare assistants are vital members of the healthcare team, as their work has been

shown to improve patient comfort during hospitalization.” [Line 49-50]

Finally, the term ‘HCA’s’ within an apostrophe before the ‘s’ is a way of showing possession, as in something belonging to a healthcare assistant. However, this is not the correct usage when simply referring to healthcare assistants in the plural. For example:

“Self-efficacy, defined as a HCA’s belief in their own capabilities to perform actions and achieve outcomes, has been thought to have direct bearing on individual performance in particular contexts.” [Line 58-60]

            In summary:

            HCAs = abbreviation for multiple healthcare assistants

            Healthcare assistants = full term for the job role

            HCA’s = possessive form, not for simple plural

Point 4: Introduction, the study’s importance has been clearly stated; however, if fails to specify the differences between this study and previous ones. What sets this study apart?

Response 4: Thank you for your comment. Following the revision process, our research team revised the following content in the ‘introduction’ part to present the differences between this study and previous ones as follow:

“However, data on infections and deaths among HCAs is severely limited across most nations, highlighting a significant gap in understanding the full extent of their experiences and challenges. The ICN advocated standardizing data collection on cases and fatalities among HCAs, which is crucial to understanding and mitigating the occupational risks they face [40]. While some studies have investigated the impact of the pan-demic on healthcare workers in general, limited qualitative research has specifically explored the unique perspectives and experiences of HCAs themselves, particularly during times of crisis, like those seen during the COVID-19 pandemic [41]. This study aims to address this gap by focusing exclusively on the voices of HCAs, providing valuable insights into their resilience and self-efficacy during this unprecedented time. The findings of this study will provide a greater understanding of HCA perspectives, ad-dressing gaps in knowledge about the development of resilience and self-efficacy of HCAs in both within the localized context of Hong Kong and the broader international landscape. By delving into the specific challenges, coping mechanisms, and support systems of HCAs in Hong Kong, this study offers a unique contribution to the existing literature, which has largely overlooked this critical workforce.

The study aimed to explore HCAs’ experiences and perspectives on resilience and self-efficacy during the COVID-19 pandemic in Hong Kong. Based on the findings of this study, management of the healthcare system can identify ways to better support and empower this critical workforce during future public health crises. By under-standing the specific needs and challenges faced by HCAs, healthcare organizations can develop targeted interventions and policies to enhance their resilience and self-efficacy, ultimately strengthening the healthcare system’s response to future

pandemics or other crises.” [Line 84-106]

Point 5: Methods: In Lines 130-131. I think the hyphenation of the word “vide-oconferencing” is not quite right. If it can be edited, write “video-conferencing” which I think is easier to read.

Response 5: Thank you for your comment. Our research team revised the typo error as follows:

“All interviews were conducted in Chinese by the second author over Zoom video-

conferencing within a private, password-protected virtual space from March 2021 to

December 2021.” [Line 140-142]

Point 6:  Participants in Lines 118-120, the number of participants is stated as 25 people. Where did that number come from? How many people are in the original population? Is there a specific formula for determining the number of participants?

Response 6: Our study included 25 healthcare assistants actively caring for COVID-19 patients in Hong Kong public hospitals. This sample size is appropriate for our qualitative research methodology, which prioritizes depth of information over participant quantity. Participants were recruited through nursing personnel’s social networks, colleagues, and hospital managers. We determined the number of participants based on the principle of data saturation, a standard practice in qualitative research where interviews continue until no new themes emerge. This approach allowed us to gather rich, detailed data sufficient for comprehensive analysis and meaningful insights into the experiences of healthcare assistants during the COVID-19 pandemic. The sample size of 25 aligns with typical guidelines for qualitative studies using in-depth interviews and provide adequate to address our research questions thoroughly. [Line 130-137]

Point 7:  Data: The author used a questionnaire in their study, which was validated by experts. However, it has not been stated what theory, regulation, or standard was used as the basis for compiling the questions.

Response 7: Thank you for your comment. The study employed a qualitative approach using a validated interview question guide. An expert panel, consisting of a qualitative research scholar and a psychological consulting specialist, validated the interview questions to ensure alignment with the study aim and focus. The questions explored participants’ psychological experiences and emotional responses while caring for COVID-19 patients. Topics covered included changes in work duties and personal lives, coping strategies, insights gained, and perspectives on their role in preventing COVID-19 spread. This approach, while not based on a specific theory or regulation, was designed to capture rich, in-depth data about the healthcare assistants’ experiences during the pandemic. [Line 147-154]

Point 8: Are the methods adequately described? (x) Can be improved

Response 8: Thank you for your advice. We have carefully reviewed your comments and agree that the Methods section can be improved. Our team has revised address the concerns raised in Point 5, 6, and 7, providing clearer and more detailed information about our study methodology. We believe these changes significantly enhance the clarity and comprehensiveness of the Methods section. [Line 108-181]

Point 9:  Table 2. Participant characteristics. Pleas type and arrange the table properly.

Response 9: We appreciate your comment regarding ‘Table 2. Participant characteristics’. We assure you that the table’s format will be properly revised and arranged by our editorial colleagues to meet the journal’s formatting requirements and ensure clear presentation of the participant characteristics. [Line 191-192]

Point 10:  The “4 Discussion” can be bolded on line 452.

Response 10: Agree. Our team revised the ‘4. Discussion’ in Line 469.

Point 11:  The author has cited an excessive number of sources for a single sentence, which is unnecessary. Please review references 17-25 (lines 64-67), 17-27 (line 67-68), and 52-60 (lines 478-481).

Response 11: Thank you for your comment. We revised the references as below:

            “This potential lack of mobility and perpetual perception of being ‘bottom of the totem

pole’ can cause HCAs to experience a lack of purpose, loss of motivation, and

emotional exhaustion—all key factors contributing to burnout [17,23–25]. However,

over the past two decades, HCA roles have expanded and gained greater value

[18,26,27].” [line 64-68]

“In Western countries and Hong Kong, social media highlighted the contributions of

physicians and nurses and brought attention to the need for targeted interventions and

support measures to address their mental health burden [56–60]” [Line 495-498].

Response to Reviewer 4 Comments

Dear Respected Reviewer,

Healthcare

RE: Manuscript titled “Navigating uncertainty with compassion: healthcare assistants’ reflections on balancing COVID-19 and routine care through adversity”

Thank you for your unstinting effort to review the revised manuscript. Our research team appreciated your valuable comments. In this document, we provided our responses in a point-by-point format. We hope that our revisions to the manuscript can address your concerns satisfactorily.

This response letter was prepared with reference to a template that we downloaded from the Journal website. In that template, we noticed that all responses from the authors were highlighted in red. We thus followed the prescribed style and format.

Yours faithfully,

Manuscript no.: healthcare-3078683

Point 1: Does the introduction provide sufficient background and include all relevant references? (x) Must be improved

Response 1: Thank you for your comment. Following the revision process, our research team revised the following content in the ‘introduction’ part as follow:

“However, data on infections and deaths among HCAs is severely limited across most nations, highlighting a significant gap in understanding the full extent of their experiences and challenges. The ICN advocated standardizing data collection on cases and fatalities among HCAs, which is crucial to understanding and mitigating the occupational risks they face [40]. While some studies have investigated the impact of the pan-demic on healthcare workers in general, limited qualitative research has specifically explored the unique perspectives and experiences of HCAs themselves, particularly during times of crisis, like those seen during the COVID-19 pandemic [41]. This study aims to address this gap by focusing exclusively on the voices of HCAs, providing valuable insights into their resilience and self-efficacy during this unprecedented time. The findings of this study will provide a greater understanding of HCA perspectives, ad-dressing gaps in knowledge about the development of resilience and self-efficacy of HCAs in both within the localized context of Hong Kong and the broader international landscape. By delving into the specific challenges, coping mechanisms, and support systems of HCAs in Hong Kong, this study offers a unique contribution to the existing literature, which has largely overlooked this critical workforce.

The study aimed to explore HCAs’ experiences and perspectives on resilience and self-efficacy during the COVID-19 pandemic in Hong Kong. Based on the findings of this study, management of the healthcare system can identify ways to better support and empower this critical workforce during future public health crises. By under-standing the specific needs and challenges faced by HCAs, healthcare organizations can develop targeted interventions and policies to enhance their resilience and self-efficacy, ultimately strengthening the healthcare system’s response to future

pandemics or other crises.” [Line 84-106]

Point 2: To examine carefully the correct way to write affiliation according to the journal guidelines.

Response 2: Agree. Our research team provided correct way to write affiliation according to the guidelines:

“Alice YIP 1,*, Jeff YIP 2, Zoe TSUI 1, and Graeme Drummond SMITH 1

1 S.K. Yee School of Health Sciences, Saint Francis University, Hong Kong, China;

  [email protected] (Z.T);

2 Hong Kong Institute of Paramedicine, Hong Kong, China; [email protected]

  (J.Y.)” [Line 5-8]

Point 3: Introduction: in the introduction, the term healthcare assistants is abbreviated as “HCAs”. However, the reviewer found many inconsistencies in the writing of the abbreviation, sometimes writing “HCAs”, sometimes writing “HCA” sometimes writing “HCA’s”. Are these three abbreviations different things? Or just typos?

Response 3: Thank you for your advice. In Line 14, this is typo error ‘healthcare assistant (HCA)’ instead of healthcare assistants (HCAs). The term ‘HCAs’ is an abbreviation for ‘healthcare assistants’. When written this way, ‘HCAs’ is referring to healthcare assistants in the plural form. For example:

“However, studies suggest HCAs were frequently deputized, their skills not fully lever-

aged, and their contributions overlooked during the pandemic.” [Line 46-47]

In this case, ‘HCAs’ means multiple healthcare assistants.

The phrase ‘healthcare assistants’ is the full, unabbreviated term. It refers to medical support staff who work under the supervision of nurses or other healthcare professionals to provide basic patient care. For example:

“Healthcare assistants are vital members of the healthcare team, as their work has been

shown to improve patient comfort during hospitalization.” [Line 49-50]

Finally, the term ‘HCA’s’ within an apostrophe before the ‘s’ is a way of showing possession, as in something belonging to a healthcare assistant. However, this is not the correct usage when simply referring to healthcare assistants in the plural. For example:

“Self-efficacy, defined as a HCA’s belief in their own capabilities to perform actions and achieve outcomes, has been thought to have direct bearing on individual performance in particular contexts.” [Line 58-60]

            In summary:

            HCAs = abbreviation for multiple healthcare assistants

            Healthcare assistants = full term for the job role

            HCA’s = possessive form, not for simple plural

Point 4: Introduction, the study’s importance has been clearly stated; however, if fails to specify the differences between this study and previous ones. What sets this study apart?

Response 4: Thank you for your comment. Following the revision process, our research team revised the following content in the ‘introduction’ part to present the differences between this study and previous ones as follow:

“However, data on infections and deaths among HCAs is severely limited across most nations, highlighting a significant gap in understanding the full extent of their experiences and challenges. The ICN advocated standardizing data collection on cases and fatalities among HCAs, which is crucial to understanding and mitigating the occupational risks they face [40]. While some studies have investigated the impact of the pan-demic on healthcare workers in general, limited qualitative research has specifically explored the unique perspectives and experiences of HCAs themselves, particularly during times of crisis, like those seen during the COVID-19 pandemic [41]. This study aims to address this gap by focusing exclusively on the voices of HCAs, providing valuable insights into their resilience and self-efficacy during this unprecedented time. The findings of this study will provide a greater understanding of HCA perspectives, ad-dressing gaps in knowledge about the development of resilience and self-efficacy of HCAs in both within the localized context of Hong Kong and the broader international landscape. By delving into the specific challenges, coping mechanisms, and support systems of HCAs in Hong Kong, this study offers a unique contribution to the existing literature, which has largely overlooked this critical workforce.

The study aimed to explore HCAs’ experiences and perspectives on resilience and self-efficacy during the COVID-19 pandemic in Hong Kong. Based on the findings of this study, management of the healthcare system can identify ways to better support and empower this critical workforce during future public health crises. By under-standing the specific needs and challenges faced by HCAs, healthcare organizations can develop targeted interventions and policies to enhance their resilience and self-efficacy, ultimately strengthening the healthcare system’s response to future

pandemics or other crises.” [Line 84-106]

Point 5: Methods: In Lines 130-131. I think the hyphenation of the word “vide-oconferencing” is not quite right. If it can be edited, write “video-conferencing” which I think is easier to read.

Response 5: Thank you for your comment. Our research team revised the typo error as follows:

“All interviews were conducted in Chinese by the second author over Zoom video-

conferencing within a private, password-protected virtual space from March 2021 to

December 2021.” [Line 140-142]

Point 6:  Participants in Lines 118-120, the number of participants is stated as 25 people. Where did that number come from? How many people are in the original population? Is there a specific formula for determining the number of participants?

Response 6: Our study included 25 healthcare assistants actively caring for COVID-19 patients in Hong Kong public hospitals. This sample size is appropriate for our qualitative research methodology, which prioritizes depth of information over participant quantity. Participants were recruited through nursing personnel’s social networks, colleagues, and hospital managers. We determined the number of participants based on the principle of data saturation, a standard practice in qualitative research where interviews continue until no new themes emerge. This approach allowed us to gather rich, detailed data sufficient for comprehensive analysis and meaningful insights into the experiences of healthcare assistants during the COVID-19 pandemic. The sample size of 25 aligns with typical guidelines for qualitative studies using in-depth interviews and provide adequate to address our research questions thoroughly. [Line 130-137]

Point 7:  Data: The author used a questionnaire in their study, which was validated by experts. However, it has not been stated what theory, regulation, or standard was used as the basis for compiling the questions.

Response 7: Thank you for your comment. The study employed a qualitative approach using a validated interview question guide. An expert panel, consisting of a qualitative research scholar and a psychological consulting specialist, validated the interview questions to ensure alignment with the study aim and focus. The questions explored participants’ psychological experiences and emotional responses while caring for COVID-19 patients. Topics covered included changes in work duties and personal lives, coping strategies, insights gained, and perspectives on their role in preventing COVID-19 spread. This approach, while not based on a specific theory or regulation, was designed to capture rich, in-depth data about the healthcare assistants’ experiences during the pandemic. [Line 147-154]

Point 8: Are the methods adequately described? (x) Can be improved

Response 8: Thank you for your advice. We have carefully reviewed your comments and agree that the Methods section can be improved. Our team has revised address the concerns raised in Point 5, 6, and 7, providing clearer and more detailed information about our study methodology. We believe these changes significantly enhance the clarity and comprehensiveness of the Methods section. [Line 108-181]

Point 9:  Table 2. Participant characteristics. Pleas type and arrange the table properly.

Response 9: We appreciate your comment regarding ‘Table 2. Participant characteristics’. We assure you that the table’s format will be properly revised and arranged by our editorial colleagues to meet the journal’s formatting requirements and ensure clear presentation of the participant characteristics. [Line 191-192]

Point 10:  The “4 Discussion” can be bolded on line 452.

Response 10: Agree. Our team revised the ‘4. Discussion’ in Line 469.

Point 11:  The author has cited an excessive number of sources for a single sentence, which is unnecessary. Please review references 17-25 (lines 64-67), 17-27 (line 67-68), and 52-60 (lines 478-481).

Response 11: Thank you for your comment. We revised the references as below:

            “This potential lack of mobility and perpetual perception of being ‘bottom of the totem

pole’ can cause HCAs to experience a lack of purpose, loss of motivation, and

emotional exhaustion—all key factors contributing to burnout [17,23–25]. However,

over the past two decades, HCA roles have expanded and gained greater value

[18,26,27].” [line 64-68]

“In Western countries and Hong Kong, social media highlighted the contributions of

physicians and nurses and brought attention to the need for targeted interventions and

support measures to address their mental health burden [56–60]” [Line 495-498].

Response to Reviewer 4 Comments

Dear Respected Reviewer,

Healthcare

RE: Manuscript titled “Navigating uncertainty with compassion: healthcare assistants’ reflections on balancing COVID-19 and routine care through adversity”

Thank you for your unstinting effort to review the revised manuscript. Our research team appreciated your valuable comments. In this document, we provided our responses in a point-by-point format. We hope that our revisions to the manuscript can address your concerns satisfactorily.

This response letter was prepared with reference to a template that we downloaded from the Journal website. In that template, we noticed that all responses from the authors were highlighted in red. We thus followed the prescribed style and format.

Yours faithfully,

Manuscript no.: healthcare-3078683

Point 1: Does the introduction provide sufficient background and include all relevant references? (x) Must be improved

Response 1: Thank you for your comment. Following the revision process, our research team revised the following content in the ‘introduction’ part as follow:

“However, data on infections and deaths among HCAs is severely limited across most nations, highlighting a significant gap in understanding the full extent of their experiences and challenges. The ICN advocated standardizing data collection on cases and fatalities among HCAs, which is crucial to understanding and mitigating the occupational risks they face [40]. While some studies have investigated the impact of the pan-demic on healthcare workers in general, limited qualitative research has specifically explored the unique perspectives and experiences of HCAs themselves, particularly during times of crisis, like those seen during the COVID-19 pandemic [41]. This study aims to address this gap by focusing exclusively on the voices of HCAs, providing valuable insights into their resilience and self-efficacy during this unprecedented time. The findings of this study will provide a greater understanding of HCA perspectives, ad-dressing gaps in knowledge about the development of resilience and self-efficacy of HCAs in both within the localized context of Hong Kong and the broader international landscape. By delving into the specific challenges, coping mechanisms, and support systems of HCAs in Hong Kong, this study offers a unique contribution to the existing literature, which has largely overlooked this critical workforce.

The study aimed to explore HCAs’ experiences and perspectives on resilience and self-efficacy during the COVID-19 pandemic in Hong Kong. Based on the findings of this study, management of the healthcare system can identify ways to better support and empower this critical workforce during future public health crises. By under-standing the specific needs and challenges faced by HCAs, healthcare organizations can develop targeted interventions and policies to enhance their resilience and self-efficacy, ultimately strengthening the healthcare system’s response to future

pandemics or other crises.” [Line 84-106]

Point 2: To examine carefully the correct way to write affiliation according to the journal guidelines.

Response 2: Agree. Our research team provided correct way to write affiliation according to the guidelines:

“Alice YIP 1,*, Jeff YIP 2, Zoe TSUI 1, and Graeme Drummond SMITH 1

1 S.K. Yee School of Health Sciences, Saint Francis University, Hong Kong, China;

  [email protected] (Z.T);

2 Hong Kong Institute of Paramedicine, Hong Kong, China; [email protected]

  (J.Y.)” [Line 5-8]

Point 3: Introduction: in the introduction, the term healthcare assistants is abbreviated as “HCAs”. However, the reviewer found many inconsistencies in the writing of the abbreviation, sometimes writing “HCAs”, sometimes writing “HCA” sometimes writing “HCA’s”. Are these three abbreviations different things? Or just typos?

Response 3: Thank you for your advice. In Line 14, this is typo error ‘healthcare assistant (HCA)’ instead of healthcare assistants (HCAs). The term ‘HCAs’ is an abbreviation for ‘healthcare assistants’. When written this way, ‘HCAs’ is referring to healthcare assistants in the plural form. For example:

“However, studies suggest HCAs were frequently deputized, their skills not fully lever-

aged, and their contributions overlooked during the pandemic.” [Line 46-47]

In this case, ‘HCAs’ means multiple healthcare assistants.

The phrase ‘healthcare assistants’ is the full, unabbreviated term. It refers to medical support staff who work under the supervision of nurses or other healthcare professionals to provide basic patient care. For example:

“Healthcare assistants are vital members of the healthcare team, as their work has been

shown to improve patient comfort during hospitalization.” [Line 49-50]

Finally, the term ‘HCA’s’ within an apostrophe before the ‘s’ is a way of showing possession, as in something belonging to a healthcare assistant. However, this is not the correct usage when simply referring to healthcare assistants in the plural. For example:

“Self-efficacy, defined as a HCA’s belief in their own capabilities to perform actions and achieve outcomes, has been thought to have direct bearing on individual performance in particular contexts.” [Line 58-60]

            In summary:

            HCAs = abbreviation for multiple healthcare assistants

            Healthcare assistants = full term for the job role

            HCA’s = possessive form, not for simple plural

Point 4: Introduction, the study’s importance has been clearly stated; however, if fails to specify the differences between this study and previous ones. What sets this study apart?

Response 4: Thank you for your comment. Following the revision process, our research team revised the following content in the ‘introduction’ part to present the differences between this study and previous ones as follow:

“However, data on infections and deaths among HCAs is severely limited across most nations, highlighting a significant gap in understanding the full extent of their experiences and challenges. The ICN advocated standardizing data collection on cases and fatalities among HCAs, which is crucial to understanding and mitigating the occupational risks they face [40]. While some studies have investigated the impact of the pan-demic on healthcare workers in general, limited qualitative research has specifically explored the unique perspectives and experiences of HCAs themselves, particularly during times of crisis, like those seen during the COVID-19 pandemic [41]. This study aims to address this gap by focusing exclusively on the voices of HCAs, providing valuable insights into their resilience and self-efficacy during this unprecedented time. The findings of this study will provide a greater understanding of HCA perspectives, ad-dressing gaps in knowledge about the development of resilience and self-efficacy of HCAs in both within the localized context of Hong Kong and the broader international landscape. By delving into the specific challenges, coping mechanisms, and support systems of HCAs in Hong Kong, this study offers a unique contribution to the existing literature, which has largely overlooked this critical workforce.

The study aimed to explore HCAs’ experiences and perspectives on resilience and self-efficacy during the COVID-19 pandemic in Hong Kong. Based on the findings of this study, management of the healthcare system can identify ways to better support and empower this critical workforce during future public health crises. By under-standing the specific needs and challenges faced by HCAs, healthcare organizations can develop targeted interventions and policies to enhance their resilience and self-efficacy, ultimately strengthening the healthcare system’s response to future

pandemics or other crises.” [Line 84-106]

Point 5: Methods: In Lines 130-131. I think the hyphenation of the word “vide-oconferencing” is not quite right. If it can be edited, write “video-conferencing” which I think is easier to read.

Response 5: Thank you for your comment. Our research team revised the typo error as follows:

“All interviews were conducted in Chinese by the second author over Zoom video-

conferencing within a private, password-protected virtual space from March 2021 to

December 2021.” [Line 140-142]

Point 6:  Participants in Lines 118-120, the number of participants is stated as 25 people. Where did that number come from? How many people are in the original population? Is there a specific formula for determining the number of participants?

Response 6: Our study included 25 healthcare assistants actively caring for COVID-19 patients in Hong Kong public hospitals. This sample size is appropriate for our qualitative research methodology, which prioritizes depth of information over participant quantity. Participants were recruited through nursing personnel’s social networks, colleagues, and hospital managers. We determined the number of participants based on the principle of data saturation, a standard practice in qualitative research where interviews continue until no new themes emerge. This approach allowed us to gather rich, detailed data sufficient for comprehensive analysis and meaningful insights into the experiences of healthcare assistants during the COVID-19 pandemic. The sample size of 25 aligns with typical guidelines for qualitative studies using in-depth interviews and provide adequate to address our research questions thoroughly. [Line 130-137]

Point 7:  Data: The author used a questionnaire in their study, which was validated by experts. However, it has not been stated what theory, regulation, or standard was used as the basis for compiling the questions.

Response 7: Thank you for your comment. The study employed a qualitative approach using a validated interview question guide. An expert panel, consisting of a qualitative research scholar and a psychological consulting specialist, validated the interview questions to ensure alignment with the study aim and focus. The questions explored participants’ psychological experiences and emotional responses while caring for COVID-19 patients. Topics covered included changes in work duties and personal lives, coping strategies, insights gained, and perspectives on their role in preventing COVID-19 spread. This approach, while not based on a specific theory or regulation, was designed to capture rich, in-depth data about the healthcare assistants’ experiences during the pandemic. [Line 147-154]

Point 8: Are the methods adequately described? (x) Can be improved

Response 8: Thank you for your advice. We have carefully reviewed your comments and agree that the Methods section can be improved. Our team has revised address the concerns raised in Point 5, 6, and 7, providing clearer and more detailed information about our study methodology. We believe these changes significantly enhance the clarity and comprehensiveness of the Methods section. [Line 108-181]

Point 9:  Table 2. Participant characteristics. Pleas type and arrange the table properly.

Response 9: We appreciate your comment regarding ‘Table 2. Participant characteristics’. We assure you that the table’s format will be properly revised and arranged by our editorial colleagues to meet the journal’s formatting requirements and ensure clear presentation of the participant characteristics. [Line 191-192]

Point 10:  The “4 Discussion” can be bolded on line 452.

Response 10: Agree. Our team revised the ‘4. Discussion’ in Line 469.

Point 11:  The author has cited an excessive number of sources for a single sentence, which is unnecessary. Please review references 17-25 (lines 64-67), 17-27 (line 67-68), and 52-60 (lines 478-481).

Response 11: Thank you for your comment. We revised the references as below:

            “This potential lack of mobility and perpetual perception of being ‘bottom of the totem

pole’ can cause HCAs to experience a lack of purpose, loss of motivation, and

emotional exhaustion—all key factors contributing to burnout [17,23–25]. However,

over the past two decades, HCA roles have expanded and gained greater value

[18,26,27].” [line 64-68]

“In Western countries and Hong Kong, social media highlighted the contributions of

physicians and nurses and brought attention to the need for targeted interventions and

support measures to address their mental health burden [56–60]” [Line 495-498].

Response to Reviewer 4 Comments

Dear Respected Reviewer,

Healthcare

RE: Manuscript titled “Navigating uncertainty with compassion: healthcare assistants’ reflections on balancing COVID-19 and routine care through adversity”

Thank you for your unstinting effort to review the revised manuscript. Our research team appreciated your valuable comments. In this document, we provided our responses in a point-by-point format. We hope that our revisions to the manuscript can address your concerns satisfactorily.

This response letter was prepared with reference to a template that we downloaded from the Journal website. In that template, we noticed that all responses from the authors were highlighted in red. We thus followed the prescribed style and format.

Yours faithfully,

Manuscript no.: healthcare-3078683

Point 1: Does the introduction provide sufficient background and include all relevant references? (x) Must be improved

Response 1: Thank you for your comment. Following the revision process, our research team revised the following content in the ‘introduction’ part as follow:

“However, data on infections and deaths among HCAs is severely limited across most nations, highlighting a significant gap in understanding the full extent of their experiences and challenges. The ICN advocated standardizing data collection on cases and fatalities among HCAs, which is crucial to understanding and mitigating the occupational risks they face [40]. While some studies have investigated the impact of the pan-demic on healthcare workers in general, limited qualitative research has specifically explored the unique perspectives and experiences of HCAs themselves, particularly during times of crisis, like those seen during the COVID-19 pandemic [41]. This study aims to address this gap by focusing exclusively on the voices of HCAs, providing valuable insights into their resilience and self-efficacy during this unprecedented time. The findings of this study will provide a greater understanding of HCA perspectives, ad-dressing gaps in knowledge about the development of resilience and self-efficacy of HCAs in both within the localized context of Hong Kong and the broader international landscape. By delving into the specific challenges, coping mechanisms, and support systems of HCAs in Hong Kong, this study offers a unique contribution to the existing literature, which has largely overlooked this critical workforce.

The study aimed to explore HCAs’ experiences and perspectives on resilience and self-efficacy during the COVID-19 pandemic in Hong Kong. Based on the findings of this study, management of the healthcare system can identify ways to better support and empower this critical workforce during future public health crises. By under-standing the specific needs and challenges faced by HCAs, healthcare organizations can develop targeted interventions and policies to enhance their resilience and self-efficacy, ultimately strengthening the healthcare system’s response to future

pandemics or other crises.” [Line 84-106]

Point 2: To examine carefully the correct way to write affiliation according to the journal guidelines.

Response 2: Agree. Our research team provided correct way to write affiliation according to the guidelines:

“Alice YIP 1,*, Jeff YIP 2, Zoe TSUI 1, and Graeme Drummond SMITH 1

1 S.K. Yee School of Health Sciences, Saint Francis University, Hong Kong, China;

  [email protected] (Z.T);

2 Hong Kong Institute of Paramedicine, Hong Kong, China; [email protected]

  (J.Y.)” [Line 5-8]

Point 3: Introduction: in the introduction, the term healthcare assistants is abbreviated as “HCAs”. However, the reviewer found many inconsistencies in the writing of the abbreviation, sometimes writing “HCAs”, sometimes writing “HCA” sometimes writing “HCA’s”. Are these three abbreviations different things? Or just typos?

Response 3: Thank you for your advice. In Line 14, this is typo error ‘healthcare assistant (HCA)’ instead of healthcare assistants (HCAs). The term ‘HCAs’ is an abbreviation for ‘healthcare assistants’. When written this way, ‘HCAs’ is referring to healthcare assistants in the plural form. For example:

“However, studies suggest HCAs were frequently deputized, their skills not fully lever-

aged, and their contributions overlooked during the pandemic.” [Line 46-47]

In this case, ‘HCAs’ means multiple healthcare assistants.

The phrase ‘healthcare assistants’ is the full, unabbreviated term. It refers to medical support staff who work under the supervision of nurses or other healthcare professionals to provide basic patient care. For example:

“Healthcare assistants are vital members of the healthcare team, as their work has been

shown to improve patient comfort during hospitalization.” [Line 49-50]

Finally, the term ‘HCA’s’ within an apostrophe before the ‘s’ is a way of showing possession, as in something belonging to a healthcare assistant. However, this is not the correct usage when simply referring to healthcare assistants in the plural. For example:

“Self-efficacy, defined as a HCA’s belief in their own capabilities to perform actions and achieve outcomes, has been thought to have direct bearing on individual performance in particular contexts.” [Line 58-60]

            In summary:

            HCAs = abbreviation for multiple healthcare assistants

            Healthcare assistants = full term for the job role

            HCA’s = possessive form, not for simple plural

Point 4: Introduction, the study’s importance has been clearly stated; however, if fails to specify the differences between this study and previous ones. What sets this study apart?

Response 4: Thank you for your comment. Following the revision process, our research team revised the following content in the ‘introduction’ part to present the differences between this study and previous ones as follow:

“However, data on infections and deaths among HCAs is severely limited across most nations, highlighting a significant gap in understanding the full extent of their experiences and challenges. The ICN advocated standardizing data collection on cases and fatalities among HCAs, which is crucial to understanding and mitigating the occupational risks they face [40]. While some studies have investigated the impact of the pan-demic on healthcare workers in general, limited qualitative research has specifically explored the unique perspectives and experiences of HCAs themselves, particularly during times of crisis, like those seen during the COVID-19 pandemic [41]. This study aims to address this gap by focusing exclusively on the voices of HCAs, providing valuable insights into their resilience and self-efficacy during this unprecedented time. The findings of this study will provide a greater understanding of HCA perspectives, ad-dressing gaps in knowledge about the development of resilience and self-efficacy of HCAs in both within the localized context of Hong Kong and the broader international landscape. By delving into the specific challenges, coping mechanisms, and support systems of HCAs in Hong Kong, this study offers a unique contribution to the existing literature, which has largely overlooked this critical workforce.

The study aimed to explore HCAs’ experiences and perspectives on resilience and self-efficacy during the COVID-19 pandemic in Hong Kong. Based on the findings of this study, management of the healthcare system can identify ways to better support and empower this critical workforce during future public health crises. By under-standing the specific needs and challenges faced by HCAs, healthcare organizations can develop targeted interventions and policies to enhance their resilience and self-efficacy, ultimately strengthening the healthcare system’s response to future

pandemics or other crises.” [Line 84-106]

Point 5: Methods: In Lines 130-131. I think the hyphenation of the word “vide-oconferencing” is not quite right. If it can be edited, write “video-conferencing” which I think is easier to read.

Response 5: Thank you for your comment. Our research team revised the typo error as follows:

“All interviews were conducted in Chinese by the second author over Zoom video-

conferencing within a private, password-protected virtual space from March 2021 to

December 2021.” [Line 140-142]

Point 6:  Participants in Lines 118-120, the number of participants is stated as 25 people. Where did that number come from? How many people are in the original population? Is there a specific formula for determining the number of participants?

Response 6: Our study included 25 healthcare assistants actively caring for COVID-19 patients in Hong Kong public hospitals. This sample size is appropriate for our qualitative research methodology, which prioritizes depth of information over participant quantity. Participants were recruited through nursing personnel’s social networks, colleagues, and hospital managers. We determined the number of participants based on the principle of data saturation, a standard practice in qualitative research where interviews continue until no new themes emerge. This approach allowed us to gather rich, detailed data sufficient for comprehensive analysis and meaningful insights into the experiences of healthcare assistants during the COVID-19 pandemic. The sample size of 25 aligns with typical guidelines for qualitative studies using in-depth interviews and provide adequate to address our research questions thoroughly. [Line 130-137]

Point 7:  Data: The author used a questionnaire in their study, which was validated by experts. However, it has not been stated what theory, regulation, or standard was used as the basis for compiling the questions.

Response 7: Thank you for your comment. The study employed a qualitative approach using a validated interview question guide. An expert panel, consisting of a qualitative research scholar and a psychological consulting specialist, validated the interview questions to ensure alignment with the study aim and focus. The questions explored participants’ psychological experiences and emotional responses while caring for COVID-19 patients. Topics covered included changes in work duties and personal lives, coping strategies, insights gained, and perspectives on their role in preventing COVID-19 spread. This approach, while not based on a specific theory or regulation, was designed to capture rich, in-depth data about the healthcare assistants’ experiences during the pandemic. [Line 147-154]

Point 8: Are the methods adequately described? (x) Can be improved

Response 8: Thank you for your advice. We have carefully reviewed your comments and agree that the Methods section can be improved. Our team has revised address the concerns raised in Point 5, 6, and 7, providing clearer and more detailed information about our study methodology. We believe these changes significantly enhance the clarity and comprehensiveness of the Methods section. [Line 108-181]

Point 9:  Table 2. Participant characteristics. Pleas type and arrange the table properly.

Response 9: We appreciate your comment regarding ‘Table 2. Participant characteristics’. We assure you that the table’s format will be properly revised and arranged by our editorial colleagues to meet the journal’s formatting requirements and ensure clear presentation of the participant characteristics. [Line 191-192]

Point 10:  The “4 Discussion” can be bolded on line 452.

Response 10: Agree. Our team revised the ‘4. Discussion’ in Line 469.

Point 11:  The author has cited an excessive number of sources for a single sentence, which is unnecessary. Please review references 17-25 (lines 64-67), 17-27 (line 67-68), and 52-60 (lines 478-481).

Response 11: Thank you for your comment. We revised the references as below:

            “This potential lack of mobility and perpetual perception of being ‘bottom of the totem

pole’ can cause HCAs to experience a lack of purpose, loss of motivation, and

emotional exhaustion—all key factors contributing to burnout [17,23–25]. However,

over the past two decades, HCA roles have expanded and gained greater value

[18,26,27].” [line 64-68]

“In Western countries and Hong Kong, social media highlighted the contributions of

physicians and nurses and brought attention to the need for targeted interventions and

support measures to address their mental health burden [56–60]” [Line 495-498].

This manuscript is a resubmission of an earlier submission. The following is a list of the peer review reports and author responses from that submission.

Round 1

Reviewer 1 Report

Comments and Suggestions for Authors

Dear editors and authors,

I would like to express my gratitude for the opportunity to review the article "Navigating uncertainty with compassion: healthcare assistants’ reflections on balancing COVID-19 and routine care through adversity".

I fully agree with the authors regarding the importance of studying the autonomy and self-efficacy of healthcare assistants during the COVID-19 pandemic. This research is crucial not only for improving responses to future health crises, but also for promoting the emotional well-being of healthcare professionals, ensuring the quality of patient care, and contributing to the continuous professional development of these individuals. Furthermore, there is a gap in the literature regarding these professionals in this specific context.

I would like to highlight two points for your consideration:

 1. Introduction

I propose that the terms "autonomy" and "self-efficacy" be clearly defined in the introduction to provide a solid foundation for the readers.

 2.4. Analysis strategy

Regarding the analysis strategy, I suggest including the Cohen's Kappa coefficient (k), which will enrich the study by providing a numerical measure of agreement on the identified themes.

The study meets the criteria of the COREQ checklist for qualitative research reports. Overall, it provides a solid bibliographic foundation, an appropriate methodology, and results well contextualized with the existing literature. Identifying limitations and proposing future studies are essential for advancing knowledge in this area.

Therefore, my decision is that the manuscript only requires minor adjustments before being considered ready for publication.

Thank you very much.

Best regards,

Author Response

Dear Respected Reviewer,

Healthcare

RE: Manuscript titled “Navigating uncertainty with compassion: healthcare assistants’ reflections on balancing COVID-19 and routine care through adversity”

Thank you for your unstinting effort to review the revised manuscript. Our research team appreciated your valuable comments. In this document, we provided our responses in a point-by-point format. We hope that our revisions to the manuscript can address your concerns satisfactorily.

This response letter was prepared with reference to a template that we downloaded from the Journal website. In that template, we noticed that all responses from the authors were highlighted in red. We thus followed the prescribed style and format.

Yours faithfully,

Manuscript no.: healthcare-2989413

Point 1: Does the introduction provide sufficient background and include all relevant references? (x) Can be improved

Responses 1: Thank you for your comment. Following the revision process, our research team consulted two senior editors who have more than 10 years of experience in academic editing. They independently reviewed the revised manuscript and corrected the language and grammatical errors. [Line 29-98]

Point 2: Are the methods adequately described? (x) Can be improved

Response 2: Thank you for your advice. After a careful review of the original manuscript, we recognized that the analysis process moved from the original format could be presented. And added the references from your advice. [Line 102-113, Line 144-169]

Point 3: Propose that the terms “autonomy” and “self-efficacy” be clearly defined in the introduction to provide a solid foundation for the readers.

Response 3: Agree. Our research team provided a more detailed description about autonomy and self-efficacy:

‘Autonomy allows HCAs to apply independent judgement within their scope of practice while working together interdependently with other healthcare providers. By allowing HCAs to fully achieve of their competencies without unnecessary practice restrictions, autonomy fosters optimal care delivery and breaks up impairments to effective practice [19,20]. Self-efficacy defined as HCA’s belief in their own capabilities to perform actions and achieve outcomes, has been extensively utilized for self-appraisal of communication competencies. This is attributable to its claimed direct bearing on individual performance in particular contexts, given the fluctuations in conduct that self-efficacy can potentially catalyze [21,22].’ [Line 54-62]

Point 4: Regarding the analysis strategy, I suggest including the Cohen’s Kappa coefficient (k), which will enrich the study by providing a numerical measure of agreement on the identified themes.

Response 4: Thank you for your advice. To ensure inter- and intra-reliability of the qualitative analysis, several steps were taken. First, as stated in Line 154 to Line 169, NVivo software was used for coding and analysis. Second, back-translation of interview transcripts was done by two independent translators - one researcher (T.Z.) and one professional translator - to ensure semantic equivalence (Morse, 2015; Tracy, 2019). Third, all transcription and initial coding was done independently by team members. Finally, any discrepancies in coding were resolved through discussion between the researchers until agreement was reached on the findings and findings of the analysis. [Line 154-169]

References:

Morse, J. M. (2015). Critical analysis of strategies for determining rigor in          qualitative inquiry. Qualitative Health Research, 25(9), 1212-1222.

Tracy, S. J. (2019). Qualitative research methods: Collecting evidence, crafting analysis, communicating impact. John Wiley & Son.

Reviewer 2 Report

Comments and Suggestions for Authors

Thank you for allowing me to review this article of significant importance which focuses on: Navigating uncertainty with compassion: healthcare assistants’ reflections on balancing COVID-19 and routine care through adversity

The title is clear and coherent with the aim and content of the article.

Abstract

The abstract is clear and describes all the essential aspects of the study. The use of acronyms is not recommended in the abstract, although they are few and do not affect the understanding of the text presented. 

Keywords

The keywords chosen are relevant to the study, but do not correspond to the indexed terms or Mesh. They are separated by commas and semicolons, including the OR term. This should be reviewed. The choice of key terms in natural language can make it difficult to present your article when searched in the database using indexed terms, reducing its recognition and dissemination.

Introduction

In the introduction, I recommend that don't start the sentence with a quote. I suggest rewording lines 29-30 and putting the World Health Organisation at the end of the sentence. This small change in writing demonstrates appropriation of the text and respects the original author's ownership of the data.

Line 53 add the reference.

Line 67-71 add the reference.

Line 84-87, first the objective of the study, then what we're going to get out of it.

The concepts of autonomy and self-efficacy are not described in the introduction, but it is important to understand them in order to judge whether the results obtained meet the objective of this work. 

Methods

It is crucial to take into account that the sampling technique utilized in this study could impede the generalizability of the results. The recruitment of participants via social media and networks may introduce bias since it may not be representative of all healthcare assistants (HCAs) in Hong Kong.

Furthermore, there is a concern regarding the absence of a clear explanation on how the researchers established data saturation.

Results

Including direct quotes from participants is a constructive way to enrich the narrative, as it adds a personal element to the reported findings, and further bolsters their credibility.

Discussion

The authors in their writing referred to their previous experiences with SARS. However, it would be helpful if they could provide more specific details on how these past experiences have influenced current practices, instead of just making general statements.

The section might improve by linking back more explicitly to each of the study's objectives or questions to show how they were addressed or resolved through the findings – lines 542-546.

References

The references used are current and relevant to the type of study presented.

I want to congratulate the authors on this paper, and I am confident that with a few changes, it will be able to be published.

Author Response

Response to Reviewer 2 Comments

Dear Respected Reviewer,

Healthcare

RE: Manuscript titled “Navigating uncertainty with compassion: healthcare assistants’ reflections on balancing COVID-19 and routine care through adversity”

Thank you for your unstinting effort to review the revised manuscript. Our research team appreciated your valuable comments. In this document, we provided our responses in a point-by-point format. We hope that our revisions to the manuscript can address your concerns satisfactorily.

This response letter was prepared with reference to a template that we downloaded from the Journal website. In that template, we noticed that all responses from the authors were highlighted in red. We thus followed the prescribed style and format.

Yours faithfully,

Manuscript no.: healthcare-2989413

Point 1: Does the introduction provide sufficient background and include all relevant references? (x) Can be improved

Responses 1: Thank you for your comment. Following the revision process, our research team consulted two senior editors who have more than 10 years of experience in academic editing. They independently reviewed the revised manuscript and corrected the language and grammatical errors. [Line 29-98]

Point 2: Are the methods adequately described? (x) Can be improved

Response 2: Thank you for your advice. After a careful review of the original manuscript, we recognized that the analysis process moved from the original format could be presented. And added the references from your advice. [Line 102-113, Line 144-147]

Point 3: The use of acronyms is not recommended in the abstract.

Response 3: Thank you for your advice. After a careful review of the original abstract, we modified the acronyms in the abstract:

‘Abstract: The coronavirus disease 2019 pandemic created unprecedented challenges for healthcare systems around the world. Healthcare assistants played a vital role in the provision of frontline patient care during this crisis. Despite their important contribution, there exists limited research that specifically examines the healthcare assistant’s experiences and perspectives of care provision during the COVID pandemic. This study explored healthcare assistants’ caring experiences and perspectives on autonomy and self-efficacy during the COVID-19 pandemic in Hong Kong. A qualitative descriptive study with semi-structured interviews were conducted with 25 healthcare assistants from public hospitals. Interview recordings were analyzed using thematic analysis. Five main themes emerged from the data: frontline reinforcement: supporting healthcare assistants through resourcing and education amidst the COVID-19 crisis, confronting uncertainty: building personal fortitude in the face of the COVID-19 pandemic, fostering collective resilience through shared support, cultivating self-efficacy through adaptive growth, and transformation through adversity: the evolution of frontline providers during crisis. The findings advocate enhancements autonomy and self-efficacy of healthcare assistant’s; this may potentially strengthen healthcare system preparedness for navigating unpredictable challenges in the future.’ [Line 10-24]

Point 4: The keywords chosen are relevant to the study, but do not correspond to the indexed terms or Mesh. They are separated by commas and semicolons, including the OR term. This should be reviewed.

Response 4: Agree. Our research team revised the keywords:

‘Keywords: care, COVID-19, coronavirus disease 2019 pandemic, autonomy, self-

efficacy, healthcare assistants, workforce issues’ [Line 25-26]

Point 5: In the introduction, I recommend that don’t start the sentence with a quote. I suggest rewording lines 29-30 and putting the World Health Organization at the end of the sentence. This small change in writing demonstrates appropriation of the text and respects the original author’s ownership of the data.

Response 5: Thank you for your advice. Our research team rephrase the sentence as follow:

‘The coronavirus disease 2019 (COVID-19) pandemic has been responsible for over 664 million confirmed cases and 6.7 million deaths worldwide, according to the World Health Organization [1].’ [Line 29-31]

Point 6: Line 53 add the reference.

Response 6: Thank you for your advice. After a careful review of the original manuscript, we add the reference [17,18]

They provide hospital-based patients with physical care while giving psychological and existential support, acting as observant monitors, and quickly alerting clinical colleagues about health status changes or decline [17,18]. [Line 51-53]

17. Travers, J.L.; Hirschman, K.B.; Naylor, M.D. Adapting Andersen’s expanded behavioral model of health services use to include older adults receiving long-term services and supports. BMC Geriatr. 2020, 20, 1– DOI:10.1186/s12877-019-1405-7.

18. Travers, J.L.; Schroeder, K.; Norful, A.A.; Aliyu, S. The influence of empowered work environments on the psychological experiences of nursing assistants during COVID-19: A qualitative study. BMC Nurs. 2020, 19, 1–12. DOI:10.1186/s12912-020-00489-9

Point 7: Line 67-71 add the reference.

Response 7: Thank you for your advice. After a careful review of the original manuscript, we add the references [17,18,37]

During the pandemic, which rapidly transformed healthcare operations, having a strong and skilled HCA workforce was critical for healthcare systems to meet escalat-ing care demands. Health Care Assistants, with greater experience and capabilities were better equipped to adapt to quickly evolving protocols, taking on extended re-sponsibilities, and providing high-quality patient care during intensely stressful and uncertain conditions [17,18,37]. [Line 75-80]

17. Travers, J.L.; Hirschman, K.B.; Naylor, M.D. Adapting Andersen’s expanded behavioral model of health services use to include older adults receiving long-term services and supports. BMC Geriatr. 2020, 20, 1–16. DOI:10.1186/s12877-019-1405-7.

18. Travers, J.L.; Schroeder, K.; Norful, A.A.; Aliyu, S. The influence of empowered work environments on the psychological experiences of nursing assistants during COVID-19: A qualitative study. BMC Nurs. 2020, 19, 1–12. DOI:10.1186/s12912-020-00489-9

37. Hagerman, H.; Högberg, H.; Skytt, B.; Wadensten, B.; Engström, M. Empowerment and performance of managers and subordinates in elderly care: A longitudinal and multilevel study. J Nurs Manag. 2017, 25, 647–656. DOI:10.1111/jonm.12504

Point 8: Line 84-87, first the objective of the study, then what we’re going to get out of it.

Response 8: Thank you for your advice. After a careful review of the original manuscript, we revise the further point of view:

‘The findings of this study will provide a greater understanding of HCA perspectives, addressing gaps in knowledge both within the localized context of Hong Kong and the broader international landscape. This study aimed to explore HCAs’ experiences and perspectives on autonomy and self-efficacy during the COVID-19 pandemic in Hong Kong. Therefore, management of healthcare system can identify ways to better support and empower this critical workforce during further public health crises.’ [Line 92-98]

Point 9: The concepts of autonomy and self-efficacy are not described in the introduction, but it is important to understand them in order to judge whether the results obtained meeting the objective of this work.

Response 9: Agree. Our research team provided a more detailed description about autonomy and self-efficacy:

‘Autonomy allows HCAs to apply independent judgement within their scope of practice while working together interdependently with other healthcare providers. By allowing HCAs to fully achieve of their competencies without unnecessary practice restrictions, autonomy fosters optimal care delivery and breaks up impairments to effective practice [19,20]. Self-efficacy defined as HCA’s belief in their own capabilities to perform actions and achieve outcomes, has been extensively utilized for self-appraisal of communication competencies. This is attributable to its claimed direct bearing on individual performance in particular contexts, given the fluctuations in conduct that self-efficacy can potentially catalyze [21,22].’ [Line 54-62]

Point 10: It is crucial to take into account that the sampling technique utilized in this study could impede the generalizability of the results. The recruitment of participants via social media and networks may introduce bias since it may not be representative of all healthcare assistants (HCAs) in Hong Kong.

Response 10: Thank you for your advice. Our research team also concerns this issue and the explanation presented in the limitation section. [Line 589-601]

Point 11: There is a concern regarding the absence of a clear explanation on how the researchers established data saturation.

Response 11: Thank you for your advice. After a careful review of the original manuscript, we revised a clear explanation on how to establish data saturation:

‘Several strategies were established in this study to enable trustworthiness of the findings, including member checking during interviews to confirm interpretations with HCAs, auditing of the data analysis process, captivating in reflexive discussion among the researchers to assess interpretive decisions, and purposeful sampling of HCAs from multiple units to capture an extensive of perspectives and experiences [50,51]. The final themes were derived through consensus and are detailed in the results.’ [Line 163-169]

Point 12: Including direct quotes from participants is a constructive way to enrich the narrative, as it adds a personal element to the reported findings, and further bolsters their credibility.

Response 12: Agree. Our research team provided a direct quote from participants with personal element in the result section [Line 209-460]

Point 13: The authors in their writing referred to their previous experiences with SARS. However, it would be helpful if they could provide more specific details on how these past experiences have influenced current practices, instead of just making general statements. The section might improve by linking back more explicitly to each of the study’s objectives or questions to show how they were addressed or resolved through the findings.

Response 13: Thank you for your advice. After a careful review of the original manuscript, we revised more explanation and link with the objective of the study:

‘Regarding essential upgrades were made including mandated staff training for PPE, purpose-built isolation wards for speedy containment. Healthcare institutions can address this stress by providing mental health services, encouraging open communication, and maintaining manageable workloads [70]. Nonetheless, the global span and longevity of the COVID-19 pandemic posed distinctive challenges. A comprehensive comparative study could illuminate these contrasts, guiding the development of effective strategies to assist HCAs in future pandemics. This study helps to investigate the firsthand experiences of HCAs’ personnel in Hong Kong lends critical perspectives that shine a light on the specific realities they face and enriches global understanding on readying for future crises by furnishing an intricate comprehension of their challenges as well as pinpointing vital lessons relevant internationally.’ [Line 541-552]

Reviewer 3 Report

Comments and Suggestions for Authors

First of all, I am grateful for the opportunity to review this manuscript. The following are a series of recommendations:

The design discusses the use of semi-structured interviews, which are data collection techniques, and the application of the COREQ checklist as a criterion of rigor, and both issues should appear in these sections. The design should indicate the theoretical-methodological orientation used for its realization.

Why do you consider as inclusion criteria at least three months of clinical exposure to individuals diagnosed with Covid-19, including continuous and intermittent periods of active care?

What type of sampling was performed?

What were the bioethical principles of this study?

What were the rigorous criteria for the study?

Has this study been evaluated by an Ethics Committee?

The first paragraph of the discussion should be revised, since it is much more like a conclusion than a search for similarities and differences with the results obtained in other studies. Also, it is commented that a series of recommendations were made after the study was carried out and this issue should be indicated in the applicability of the work carried out, at the end of the discussion.

Author Response

Response to Reviewer 3 Comments

Dear Respected Reviewer,

Healthcare

RE: Manuscript titled “Navigating uncertainty with compassion: healthcare assistants’ reflections on balancing COVID-19 and routine care through adversity”

Thank you for your unstinting effort to review the revised manuscript. Our research team appreciated your valuable comments. In this document, we provided our responses in a point-by-point format. We hope that our revisions to the manuscript can address your concerns satisfactorily.

This response letter was prepared with reference to a template that we downloaded from the Journal website. In that template, we noticed that all responses from the authors were highlighted in red. We thus followed the prescribed style and format.

Yours faithfully,

Manuscript no.: healthcare-2989413

Point 1: Are the methods adequately described? (x) Must be improved

Response 1: Thank you for your advice. After a careful review of the original manuscript, we recognized that the methods must be presented more clearly. [Line 103-115, Line 146-149, Line 165-171]

Point 2: The design should indicate the theoretical-methodological orientation used for its realization.

Response 2: Thank you for your advice. After a careful review of the original abstract, we presented the design more clearly:

‘This descriptive qualitative study adopted an eclectic, flexible approach grounded in constructivist inquiry [42,43]. The aim was to generate rich, realistic findings reflecting the experiences of HCAs caring for COVID-19 patients during the pandemic. The researchers opted for an adaptive methodology without stringent rules to optimize data collection and analysis based on the specific research goals. This enabled tailored methods to prioritize eliciting detailed accounts from participants. This study involved semi-structured interviews with HCA participants and thematic analysis of the inter-view transcripts (HCAs in their own words) to systematically identify major themes related to their perspectives and experiences during the COVID-19 pandemic [44]. The Consolidated Criteria for Reporting Qualitative Research (COREQ) checklist was applied in this study as a framework to ensure rigor in reporting of the qualitative methods and findings [45].’ [Line 104-115]

Point 3: Why do you consider as inclusion criteria at least three months of clinical exposure to individuals diagnosed with COVID-19, including continuous and intermittent periods of active care?

Response 3: Thank you for your advice. Our research team required at least three months of clinical exposure to COVID-19 patients as an inclusion criterion for the HCAs participating in this study in order to capture perspectives from assistants with substantial experience across multiple phases of the pandemic in 2021. By recruiting participants from Marc to December 2021 and requiring three months minimum exposure, our research team could include HCAs who cared for patients during the earlier and later parts of the year, when the pandemic and protocols may have shifted. This ensured the study included participants who had adequate long-term experience beyond just the initial adjustment period in 2021 and who could provide informed insights on caring for COVID-19 patients throughout the course of the year.

Point 4: What type of sampling was performed?

Response 4: This study applied purposive sampling.

‘Employing a purposive sampling approach, 25 HCAs from public hospitals in Hong Kong who actively attended to COVID-19 patients were selected for study participation.’ [Line 122-123]

Point 5: What were the bioethical principles of this study?

Response 5: Principles of autonomy and self-efficacy were the bioethical principles of this study.

Point 6: What were the rigorous criteria for the study?

Response 6: The rigorous criteria for the study presented as below:

‘Following data collection process, thematic analysis was performed using NVivo software (NVivo Version 12, QSR International) without delay to analyze the inter-view transcripts [46]. Two researchers (Y.A. and T.Z.) created verbatim transcripts of the audio recordings then translated the transcripts into English. Back-translation was conducted by another researcher (T.Z.) and a professional translator to ensure semantic equivalence. The first author (Y.A.), as primary coder, initiated coding and evaluated preliminary outcomes at weekly team meetings where feedback was provided on coding and categorization until consensus was reached and the codebook finalized. Additional coding was then conducted. To ensure accuracy, researcher (T.Z.) independently double-coded the transcripts. The team had prior experience with qualitative analysis [17,18,24,47-49]. Several strategies were established in this study to enable trustworthiness of the findings, including member checking during interviews to confirm interpretations with HCAs, auditing of the data analysis process, captivating in reflexive discussion among the researchers to assess interpretive decisions, and purposeful sampling of HCAs from multiple units to capture an extensive of perspectives and experiences [50,51]. The final themes were derived through consensus and are detailed in the results.’ [Line 156-171]

Point 7: Has this study been evaluated by an Ethics Committee?

Response 7: Yes. This study granted an ethical approval from the local academic institute in Hong Kong stated as below:

‘A local academic institution’s research and ethical committee granted ethical approval (HRE210101). Participants voluntary participant in this investigation provided their written informed consent prior to study engagement. They were informed that their experiences and the content of their interviews would be shared in research dissemination, and their anonymity guaranteed to protect their personal identity.’ [Line 116-120]

Institutional Review Board Statement: The study was conducted in accordance with the Declaration of Helsinki and was approved by the Research and Ethics Committee of Caritas Institute of Higher Education (HRE210101).  Informed consent was taken from participants to participate in the study. In addition, all participants were assured that their shared experience and interview content would be reported in international journals anonymously.’ [Line 657-661]

Point 8: The first paragraph of the discussion should be revised, since it is much more like a conclusion than a search for similarities and differences with the results obtained in other studies. Also, it is commented that a series of recommendations were made after the study was carried out and this issue should be indicated in the applicability of the work carried out, at the end of the discussion.

Response 8: Thank you for your advice. After a careful review of the original manuscript, we revised the first paragraph at the end of the discussion section.

HCAs are integral members of healthcare workforce who provide crucial frontline care, as seen during public health crisis like the COVID-19 pandemic [75]. However, limited research exists examining the firsthand perspectives and experiences of HCAs themselves regarding care provision during such events [58]. This study helps address that gap through in-depth interviews exploring the reflections of participants in Hong Kong on delivering care amidst COVID-19. Notably, Hong Kong has previous experience responding to an infectious outbreak during the 2003 SARS pandemic [76]. Recommendations were put forth after SARS to better empower frontline healthcare workers, including ensuring adequate supplies and ongoing training. The participants in this study were able to use their SARS experience and mentors’ training during the COVID-19 pandemic. This shows that some of the recommendations made after SARS to prepare and protect frontline workers had been followed. However, more research from the HCAs perspective is needed to fully understand how much COVID-19 support came from lessons learned from SARS.’ [Line 576-589]

The recommendations presented at section 4.2 Implications for healthcare practice and policy:

‘Based on the findings, recommendations for further action encompass:

-  Conducting additional research to explore the effects of varied organizational      structures on HCA utilizing comparative or longitudinal approaches.

-  Investigating how specific empowerment strategies impact HCAs and patient        outcomes.

-  Fostering more supportive work environments to optimize HCA job 

    satisfaction and efficacy.

-  Implementing training programs to nurture HCAs with competencies for caring

    patients in unpredictable situations.

-   Ensuring adequate preparation for prospective health crises, particularly

     provisioning HCAs with requisite support and resources.’ [Line 620-630]